# Provable Membership Inference Privacy

**Zachary Izzo***                                                              *zach@nec-labs.com*
*NEC Labs America*

**Jinsung Yoon**                                                               *jinsungyoon@google.com*
*Google Cloud AI*

**Sercan Ö. Arık**                                                            *soarik@google.com*
*Google Cloud AI*

**James Zou**                                                                 *jamesz@stanford.edu*
*Department of Biomedical Data Science*
*Stanford University*

**Reviewed on OpenReview:** *https://openreview.net/forum?id=3ludyxPbb6*

## Abstract

In applications involving sensitive data, such as finance and healthcare, the necessity for preserving data privacy can be a significant barrier to machine learning model development. Differential privacy (DP) has emerged as one canonical standard for provable privacy. However, DP's strong theoretical guarantees often come at the cost of a large drop in its utility for machine learning; and DP guarantees themselves are difficult to interpret. In this work, we propose a novel privacy notion, membership inference privacy (MIP), as a step towards addressing these challenges. We give a precise characterization of the relationship between MIP and DP, and show that in some cases, MIP can be achieved using less amount of randomness compared to the amount required for guaranteeing DP, leading to smaller drop in utility. MIP guarantees are also easily interpretable in terms of the success rate of membership inference attacks in a simple random subsampling setting. As a proof of concept, we also provide a simple algorithm for guaranteeing MIP without needing to guarantee DP.

## 1 Introduction

As the popularity and efficacy of machine learning (ML) have increased, the number of domains in which ML is applied has also expanded greatly. Some of these domains, such as finance or healthcare, have ML on sensitive data which cannot be publicly shared due to regulatory or ethical concerns (Assefa et al., 2020; Office for Civil Rights, 2002). In these instances, maintaining data privacy is of paramount importance and must be considered at every stage of the ML process, from model development to deployment. Even sharing data in-house while retaining the appropriate level of privacy can be a barrier to model development (Assefa et al., 2020). During deployment, the trained model itself can leak information about the training data if proper precautions are not taken (Shokri et al., 2017; Carlini et al., 2021a).

Differential privacy (DP) (Dwork et al., 2014) has emerged as a gold standard for provable privacy in the academic literature. Training methods for DP use randomized algorithms applied on databases of points, and DP stipulates that the algorithm's random output cannot change much depending on the presence or absence of an individual point in the database. These guarantees in turn give information theoretic protection against the maximum amount of information that an adversary can obtain about any particular sample in the dataset, regardless of that adversary's prior knowledge or computational power, making DP an attractive method for guaranteeing privacy. However, DP's strong theoretical guarantees often come at the cost of a large drop in

---

*Part of this project was done while the author was an intern at Google Cloud AI.

|  | Privacy Guarantees | Noise Required | Relevant Quantities |
|---|---|---|---|
| **DP** | Stronger | Sometimes higher | Probability laws/hypothesis testing |
| **MIP (Ours)** | Weaker | Sometimes lower | Subsampling and attacker performance |

Table 1: A summary of the differences between differential privacy (DP) and membership inference privacy (MIP, our notion). DP comes with strong privacy guarantees, but these strong guarantees require large amounts of noise to be added to the algorithm. MIP is a relaxed notion of privacy, providing weaker privacy guarantees but also potentially requiring less noise and allowing for greater utility. MIP guarantees are defined in terms of simple quantities, and therefore may be easier for non-experts to interpret.

utility for many algorithms (Jayaraman and Evans, 2019). In addition, DP guarantees themselves are difficult to interpret by non-experts. There is a precise definition for what it means for an algorithm to satisfy DP with $\varepsilon = 10$, but it is not a priori clear what this definition guarantees in terms of practical questions that users could have, the most basic of which might be to ask whether an attacker can determine whether that user's information was included in the algorithm's input. These constitute challenges for adoption of DP in practice.

In this paper, we develop the theoretical underpinnings of a novel privacy notion, membership inference privacy (MIP), as a first step towards addressing these challenges. Membership inference measures privacy via a game played between the algorithm designer and an adversary or attacker. The adversary is presented with the algorithm's output and a "target" sample, which may or may not have been included in the algorithm's input set. The adversary's goal is to determine whether the target sample was included in the algorithm's input. If the adversary can succeed with probability much higher than random guessing, the algorithm must be leaking information about its input.

Membership inference is one of the simplest possible privacy violations, as the attacker only needs to infer a single bit; thus, provably protecting against membership inference provides a strong privacy guarantee. Furthermore, MIP may be easier for non-experts to interpret, as it is measured with respect to a simple quantity–namely, the maximum success rate of an attacker on the membership inference task, which requires only a simple sampling scheme to implement. In summary, our contributions are as follows:

- We propose a novel privacy notion, which we dub membership inference privacy (MIP), and study its properties.

- We characterize the relationship between MIP and differential privacy (DP). In particular, we show that DP is sufficient to certify MIP and quantify the correspondence.

- We demonstrate that in some cases, MIP can be certified using less randomness than that required for certifying DP. In other words, while DP is sufficient for certifying MIP, it is not necessary. Thus, MIP is a weaker (but still powerful!) notion of privacy than DP, allowing for greater utility on the base task. A comparison of MIP and DP features is given in Table 1.

- We introduce a "wrapper" method for turning any base algorithm with continuous outputs into an algorithm which satisfies MIP.

## 2 Related Work

**Privacy Attacks in ML**   The study of privacy attacks has recently gained popularity in the machine learning community as the importance of data privacy has become more apparent. In a *membership inference* attack (Shokri et al., 2017), an attacker is presented with a particular sample and the output of the algorithm to be attacked. The attacker's goal is to determine whether the presented sample was included in the training data or not. If the attacker can determine the membership of the sample with a probability much greater than random guessing, this indicates that the algorithm is leaking information about its training data. Obscuring whether a given individual belongs to the private dataset is the core promise of private data sharing, and the

main reason that we focus on membership inference as the privacy measure. Membership inference attacks against predictive models have been studied extensively (Shokri et al., 2017; Baluta et al., 2022; Hu et al., 2022; Liu et al., 2022; He et al., 2022; Carlini et al., 2021a), and recent works have also developed membership inference attacks against synthetic data (Stadler et al., 2022; Chen et al., 2020).

In a reconstruction attack, the attacker is not presented with a real sample to classify as belonging to the training set or not, but rather has to *create* samples belonging to the training set based only on the algorithm's output. Reconstruction attacks have been successfully conducted against large language models (Carlini et al., 2021b). At present, these attacks require the attacker to have a great deal of auxiliary information to succeed. For our purposes, we are interested in privacy attacks to measure the privacy of an algorithm, and such a granular task may place too high burden on the attacker to accurately detect "small" amounts of privacy leakage.

In an attribute inference attack (Bun et al., 2021; Stadler et al., 2022), the attacker tries to infer a sensitive attribute from a particular sample, based on its non-sensitive attributes and the attacked algorithm outputs. It has been argued that attribute inference is the entire goal of statistical learning, and therefore should not be considered a privacy violation (Bun et al., 2021; Jayaraman and Evans, 2022).

**Differential Privacy (DP)**   DP (Dwork et al., 2014) and its variants (Mironov, 2017; Dwork and Rothblum, 2016) offer strong, information-theoretic privacy guarantees. A DP (probabilistic) algorithm is one in which the probability law of its output does not change much if one sample in its input is changed. To be precise, given two datasets $D$ and $D'$, we say that these datasets are *adjacent* (denoted $D \sim D'$) if $|D \setminus D'| = |D' \setminus D| = 1$, or equivalently if $D$ and $D'$ differ by replacing exactly one element. We say that the algorithm $\mathcal{A}$ is $(\varepsilon, \delta)$-DP if

$$\mathbb{P}(\mathcal{A}(D) \in S) \leq e^{\varepsilon}\mathbb{P}(\mathcal{A}(D') \in S) + \delta$$

for any subset $S$ of the output space, and for any two adjacent datatsets $D \sim D'$. DP has many desirable properties, such as the ability to compose DP methods or post-process the output without losing guarantees. Many simple "wrapper" methods are also available for certifying DP. Among the simplest, the Laplace mechanism, adds Laplace noise to the algorithm output. The noise level must generally depend on the *sensitivity* of the base algorithm, which measures how much a single input sample can change the algorithm's output. The method we propose in this work is similar to the Laplace mechanism, but we show that the amount of noise needed can be reduced drastically. Enforcing DP with high levels of privacy (small $\varepsilon$) often comes with sharp decreases in algorithm utility (Tao et al., 2021; Stadler et al., 2022). DP is also difficult to audit; it must be proven mathematically for a given algorithm, and checking it empirically is generally computationally intractable (Gilbert and McMillan, 2018). The difficulty of checking DP has led to widespread implementation issues (and even errors due to finite machine precision), which invalidate the guarantees of DP (Jagielski et al., 2020).

While the basic definition of DP can be difficult to interpret, equivalent "operational" definitions have been developed (Wasserman and Zhou, 2010; Kairouz et al., 2015; Nasr et al., 2021). These works show that DP can equivalently be expressed in terms of the maximum success rate on an adversary which seeks to distinguish between two adjacent datasets $D$ and $D'$, given only the output of a DP algorithm. While similar to the setting of membership inference considered in this paper at face value, there are subtle differences. In particular, in the case of membership inference, one must consider *all* datasets which could have contained the target record, and *all* datasets which do not contain the target record, and distinguish between the algorithm's output in this larger space of possibilities. This is in contrast to the characterization of DP above, which only needs to distinguish between a single pair of datasets at any given time. Humphries et al. (2023) also studied membership inference attacks against DP-SGD in the presence of data dependencies, and they used the operational definitions to bound the success rate of a MIA against a DP algorithm in a simplified membership inference setting where the whole training dataset except for the target record is known, effectively reducing the problem to the hypothesis testing formulation of DP. Our methods also make use of subsampling, and therefore bear some resemblance to the DP literature on privacy amplification via subsampling (Balle et al., 2018). These results show that the privacy level of DP algorithms can be amplified if the input is first subsampled. In the present work, subsampling is instead used for privacy *measurement*, and thus constitutes a fundamentally different use case. (Refer to Section 3.3 for more details.) The independent

work of Thudi et al. (2022) specifically applies DP to bound membership inference rates, and our results in Section 3.4 complement theirs on the relationship between membership inference and DP. However, our results show that DP is not *required* to prevent membership inference; it is merely one option, and we give alternative methods for defending against membership inference.

**Membership Inference as a Measure of Privacy**  Previous works have also sought to understand privacy guarantees via membership inference. Lee and Clifton (2012) propose a privacy notion they call *differential identifiability* in order to address the difficulty of interpreting DP guarantees. Li et al. (2013) propose a notion called *membership privacy.* Both of these notions rely on bounding the change between an attacker's prior and posterior belief on the membership of target records, conditional on observing the output of the private algorithm. As we will discuss in detail in Section 3.4, because these notions require conditional, rather than marginal, bounds on the posterior, they are also distinct from (and generally stronger than) MIP. Nasr et al. (2018) also introduce a notion of membership privacy (which is technically distinct from Li et al. (2013)), which they enforce via a min-max optimization problem. While they do provide theoretical results, these guarantees hold only assuming that the optimal membership inference adversary can be computed exactly; in practice, this must be approximated (e.g. with a neural network). Furthermore, their notion of membership privacy does not allow the user to specify the desired privacy level (i.e., something akin to the $\varepsilon$ parameter for DP or the $\eta$ parameter for MIP, which we will introduce shortly). Long et al. (2017) propose an empirical measure of privacy loss based on membership inference. Experiments confirm that their notion correlates closely with empirical privacy loss, but their heuristic does not come with provable guarantees.

## 3 Membership Inference Privacy

In this section, we will motivate and define membership inference privacy and derive several theoretical results pertaining to it. Full proofs and/or proof sketches of all propositions and theorems can be found the Appendix.

### 3.1 Notation

We make use of the following notation. We will use $\mathcal{D}$ to refer to our entire dataset, which consists of $n$ samples all of which must remain private. We will use $\mathbf{x} \in \mathcal{D}$ or $\mathbf{x}^* \in \mathcal{D}$ to refer to a particular sample. $\mathcal{D}_{\text{train}} \subseteq \mathcal{D}$ refers to a size-$k$ subset of $\mathcal{D}$. We will assume the subset is selected randomly, so $\mathcal{D}_{\text{train}}$ is a random variable. The remaining data $\mathcal{D} \setminus \mathcal{D}_{\text{train}}$ will be referred to as the holdout data. We denote by $\mathbb{D}$ the set of all size-$k$ subsets of $\mathcal{D}$ (i.e., all possible training sets), and we will use $D \in \mathbb{D}$ to refer to a particular realization of the random variable $\mathcal{D}_{\text{train}}$. Given a particular sample $\mathbf{x}^* \in \mathcal{D}$, $\mathbb{D}^{\text{in}}$ (resp. $\mathbb{D}^{\text{out}}$) will refer to sets $D \in \mathbb{D}$ for which $\mathbf{x}^* \in D$ (resp. $\mathbf{x}^* \notin D$). Lastly, we will use the notation $D \sim D'$ to denote that the datasets $D$ and $D'$ are *adjacent*, that is, they differ in exactly one element.

### 3.2 Theoretical Motivation

The implicit assumption behind the public release of any statistical algorithm–be it a generative or predictive ML model, or even the release of simple population statistics–is that it is acceptable for *statistical information about the modelled data* to be released publicly. In the context of membership inference, this poses a potential problem: if the population we are modeling is significantly different from the "larger" population, then if our algorithm's output contains any useful information whatsoever, it *should* be possible for an attacker to infer whether a given record could have plausibly come from our training data or not.

We illustrate this concept with an example. Suppose the goal is to publish an ML model which predicts a patient's blood pressure from several biomarkers, specifically for patients who suffer from a particular chronic disease. To do this, we collect a dataset of individuals with confirmed cases of the disease, and use this data to train a linear regression model with coefficients $\hat{\theta}$. Formally, we let $\mathbf{x} \in \mathbb{R}^d$ denote the features (e.g. biomarker values), $z \in \mathbb{R}$ denote the patient's blood pressure, and $y = \mathbb{1}\{\text{patient has the chronic disease in question}\}$. In this case, the private dataset $\mathcal{D}_{\text{train}}$ contains only the patients with $y = 1$. Assume that in the general

populace, patient features are drawn from a mixture model:

$$y \sim \text{Bernoulli}(p), \qquad \mathbf{x} \sim \mathcal{N}(0, I), \qquad z | \mathbf{x}, y \sim \theta_y^\top \mathbf{x}, \qquad \theta_0 \neq \theta_1.$$

In the membership inference attack scenario, an adversary observes a data point $(\mathbf{x}^*, z^*)$ and the model $\hat{\theta}$, and tries to determine whether $(\mathbf{x}^*, z^*) \in \mathcal{D}_{\text{train}}$. If $\theta_0$ and $\theta_1$ are well-separated, then an adversary can train an effective classifier to determine the corresponding label $\mathbb{1}\{(\mathbf{x}^*, z^*) \in \mathcal{D}_{\text{train}}\}$ for $(\mathbf{x}^*, z^*)$ by checking whether $z^* \approx \hat{\theta}^\top \mathbf{x}^*$. Since only data with $y = 1$ belong to $\mathcal{D}_{\text{train}}$, this provides a signal to the adversary as to whether $\mathbf{x}^*$ could have belonged to $\mathcal{D}_{\text{train}}$ or not. The point is that in this setting, this outcome is unavoidable if $\hat{\theta}$ is to provide any utility whatsoever. In other words:

*In order to preserve utility, membership inference privacy must be measured with respect to the distribution from which the private data are drawn.*

The example above motivates the following theoretical ideal for our membership inference setting. Let $\mathcal{D} = \{\mathbf{x}_i\}_{i=1}^n$ be the private dataset and suppose that $\mathbf{x}_i \overset{\text{i.i.d.}}{\sim} \mathcal{P}$ for some probability distribution $\mathcal{P}$. (Note: Here, $\mathbf{x}^*$ corresponds to the complete datapoint $(\mathbf{x}^*, z^*)$ in the example above.) Let $\mathcal{A}$ be our (randomized) algorithm, and denote its output by $\theta = \mathcal{A}(\mathcal{D})$. We generate a test point by first drawing $y^* \sim \text{Bernoulli}(1/2)$, then

$$\mathbf{x}^* | y^* \sim y^* \text{Unif}(\mathcal{D}_{\text{train}}) + (1 - y^*)\mathcal{P},$$

i.e. $\mathbf{x}^*$ is a fresh draw from $\mathcal{P}$ or a random element of the private training data with equal probability. Let $\mathcal{I}$ denote any membership inference algorithm which takes as input $\mathbf{x}^*$ and the algorithm's output $\theta = \mathcal{A}(\mathcal{D}_{\text{train}})$. The notion of privacy we wish to enforce is that $\mathcal{I}$ cannot do much better to ascertain the membership of $\mathbf{x}^*$ than guessing randomly:

$$\mathbb{P}(\mathcal{I}(\mathbf{x}^*, \theta) = y^*) \leq 1/2 + \eta, \tag{1}$$

where the probability is taken over the randomness in $\mathcal{A}$ and $\mathcal{D}_{\text{train}}$, and ideally $\eta \ll 1/2$.

### 3.3 Approximation via the Empirical Distribution

In reality, we do not have access to the underlying distribution $\mathcal{P}$. Instead, we propose to use samples from a suitable empirical distribution to approximate fresh draws from $\mathcal{P}$.

**Definition 1** (Membership Inference Privacy (MIP)). *Given fixed $k \leq n$, let $\mathcal{D}_{\text{train}} \subseteq \mathcal{D}$ be a size-$k$ subset chosen uniformly at random from the elements in $\mathcal{D}$. For $\mathbf{x}^* \in \mathcal{D}$, let $y^* = \mathbb{1}\{\mathbf{x}^* \in \mathcal{D}_{\text{train}}\}$. An algorithm $\mathcal{A}$ is $\eta$-MIP with respect to $\mathcal{D}$ if for any identification algorithm $\mathcal{I}$ and for every $\mathbf{x}^* \in \mathcal{D}$, we have*

$$\mathbb{P}(\mathcal{I}(\mathbf{x}^*, \mathcal{A}(\mathcal{D}_{\text{train}})) = y^*) \leq \max\left\{\frac{k}{n}, 1 - \frac{k}{n}\right\} + \eta.$$

*Here, the probability is taken over the uniformly random size-$k$ subset $\mathcal{D}_{\text{train}} \subseteq \mathcal{D}$, as well as any randomness in $\mathcal{A}$ and $\mathcal{I}$.*

Definition 1 states that given the output of $\mathcal{A}$, an adversary cannot determine whether a given point was in the holdout set or training set with probability more than $\eta$ better than always guessing the a priori more likely outcome. In the remainder of the paper, we will set $k = n/2$, so that $\mathcal{A}$ is $\eta$-MIP if an attacker cannot have average accuracy greater than $(1/2 + \eta)$. This gives the largest a priori entropy for the attacker's classification task, which creates the highest ceiling on how much of an advantage an attacker can possibly gain from the algorithm's output, and consequently the most accurate measurement of privacy leakage. The choice $k = n/2$ also keeps us as close as possible to the theoretical motivation in the previous subsection. We note that analogues of all of our results apply for general $k$.

We also remark that MIP guarantees specify that the adversary cannot distinguish between points in $\mathcal{D}_{\text{train}}$ and $\mathcal{D} \setminus \mathcal{D}_{\text{train}}$; however, this does not rule out the possibility that the adversary may be able to determine whether or not a point was in the "base" dataset $\mathcal{D}$ or not. Indeed, the leakage about membership information

in $\mathcal{D}$ can be arbitrarily high. Consider, for instance, the following "algorithm": $\mathcal{A}(D) = \mathbb{1}\{D \subseteq \mathcal{D}\}$. The output of this algorithm gives the adversary no information about whether or not a given $\mathbf{x}^*$ was in $\mathcal{D}_{\text{train}}$ for any random subset of $\mathcal{D}$, and it therefore clearly satisfies MIP. However, the output very clearly reveals information about membership in $\mathcal{D}$. For this reason, in order for MIP to apply, membership in the "overall population" $\mathcal{D}$ should not be considered private, only membership in $\mathcal{D}_{\text{train}}$. This is a concrete consequence of the observation made in Section 3.2 that membership inference privacy must be measured with respect to the distribution of the private data if we are to preserve any utility. Knowledge of membership in $\mathcal{D}$ is essentially knowledge that the target $\mathbf{x}^*$ was drawn from the private data distribution.

The definition of MIP is phrased with respect to *any* classifier (whose randomness is independent of the randomness in $\mathcal{A}$; if the adversary knows the algorithm and the random seed, we have no hope of enforcing privacy). While this definition is compelling in that it shows a bound on what any attacker can hope to accomplish, the need to consider all possible attack algorithms makes it difficult to work with technically. The following proposition shows that MIP is equivalent to a simpler definition which does not need to simultaneously consider all identification algorithms $\mathcal{I}$. We state the theorem in terms of an algorithm with discrete output for simplicity, but the result readily generalizes to continuous output by replacing probability masses with densities.

**Proposition 2.** *Let* $\mathbb{A} = \text{Range}(\mathcal{A})$ *and for simplicity assume that* $\mathbb{A}$ *is discrete. Let* $\mathbb{P}(\mathcal{A}(\mathcal{D}_{\text{train}}) = A)$ *denote the probability of a given output* $A \in \mathbb{A}$ *over the randomness in* $\mathcal{D}_{\text{train}}$ *and* $\mathcal{A}$. *Then* $\mathcal{A}$ *is* $\eta$-*MIP if and only if*

$$\sum_{A \in \mathbb{A}} \Big( \max \big\{ \mathbb{P}(\mathbf{x}^* \in \mathcal{D}_{\text{train}} \mid \mathcal{A}(\mathcal{D}_{\text{train}}) = A), \mathbb{P}(\mathbf{x}^* \notin \mathcal{D}_{\text{train}} \mid \mathcal{A}(\mathcal{D}_{\text{train}}) = A) \big\} \, \mathbb{P}(\mathcal{A}(\mathcal{D}_{\text{train}}) = A) \Big) \le \frac{1}{2} + \eta.$$

*Furthermore, the optimal adversary is given by*

$$\mathcal{I}(\mathbf{x}^*, A) = \mathbb{1}\{\mathbb{P}(\mathbf{x}^* \in \mathcal{D}_{\text{train}} \mid \mathcal{A}(\mathcal{D}_{\text{train}}) = A) \ge 1/2\}.$$

Proposition 2 makes precise the intuition that the optimal attacker should guess the more likely of $\mathbf{x}^* \in \mathcal{D}_{\text{train}}$ or $\mathbf{x}^* \notin \mathcal{D}_{\text{train}}$ conditional on the output of $\mathcal{A}$. The optimal attacker's overall accuracy is then computed by marginalizing this conditional statement.

Finally, MIP also satisfies a post-processing inequality similar to the classical result in DP (Dwork et al., 2014). This states that any local functions of a MIP algorithm's output cannot degrade the privacy guarantee.

**Theorem 3.** *Suppose that* $\mathcal{A}$ *is* $\eta$-*MIP, and let* $f$ *be any (potentially randomized, with randomness independent of* $\mathcal{D}_{\text{train}}$*) function. Then* $f \circ \mathcal{A}$ *is also* $\eta$-*MIP.*

*Proof.* Let $\mathcal{I}_f$ be any membership inference algorithm for $f \circ \mathcal{A}$. A membership inference attack against $\mathcal{A}$ can be *any* binary function of the attacked point $\mathbf{x}^*$ and the algorithm output $A$. In particular, $\mathcal{I}_f(\mathbf{x}^*, f(A))$ is such a function, so it is permissible to define $\mathcal{I}_{\mathcal{A}}(\mathbf{x}^*, A) = \mathcal{I}_f(\mathbf{x}^*, f(A))$. Since $\mathcal{A}$ is $\eta$-MIP, we have

$$\frac{1}{2} + \eta \ge \mathbb{P}(\mathcal{I}_{\mathcal{A}}(\mathbf{x}^*, \mathcal{A}(\mathcal{D}_{\text{train}})) = y^*) = \mathbb{P}(\mathcal{I}_f(\mathbf{x}^*, f(\mathcal{A}(\mathcal{D}_{\text{train}}))) = y^*).$$

Thus, $f \circ \mathcal{A}$ is $\eta$-MIP by Definition 1. $\qquad\qquad\square$

For example, Theorem 3 is important for the application of MIP to generative model training – if we can guarantee that our generative model is $\eta$-MIP, then any output produced by it is $\eta$-MIP as well.

### 3.4 Relation to Differential Privacy

In this section, we make precise the relationship between MIP and the most common theoretical formulation of privacy: differential privacy (DP). Proofs of our results can be found in the Appendix. Our first theorem shows that DP is at least as strong as MIP.

**Theorem 4.** *Let $\mathcal{A}$ be $(\varepsilon, \delta)$-DP. Then $\mathcal{A}$ is $\eta$-MIP with $\eta = \delta + \frac{1-\delta}{1+e^{-\varepsilon}} - \frac{1}{2}$. Furthermore, this bound is tight, i.e. for any $\varepsilon > 0$ and $0 \leq \delta < 1$, there exists an $(\varepsilon, \delta)$-DP algorithm against which the optimal attacker has accuracy $\delta + \frac{1-\delta}{1+e^{-\varepsilon}}$.*

The proof makes use of the hypothesis testing interpretation of DP introduced by Kairouz et al. (2015), which we restate here for convenience. Let $D_0 \sim D_1$ be adjacent datasets, and let $Y = \mathcal{A}(D)$ where $D \in \{D_0, D_1\}$. We define the null and alternative hypotheses

$$H_0 \ : \ D = D_0, \qquad H_1 \ : \ D = D_1.$$

Hypothesis testing in this context reduces to defining a *rejection set $S$*: whenever $\mathcal{A}(D) \in S$, we reject the null hypothesis, otherwise if $\mathcal{A}(D) \notin S$, we fail to reject. Kairouz et al. (2015) showed that $\mathcal{A}$ is $(\varepsilon, \delta)$-DP if and only if for all adjacent datasets $D_0$ and $D_1$, and for any rejection set $S$, we have

$$\mathbb{P}(\mathcal{A}(D_0) \in S) + e^{\varepsilon} \mathbb{P}(\mathcal{A}(D_1) \notin S) \geq 1 - \delta,$$
$$e^{\varepsilon} \underbrace{\mathbb{P}(\mathcal{A}(D_0) \in S)}_{\text{Type I error}} + \underbrace{\mathbb{P}(\mathcal{A}(D_1) \notin S)}_{\text{Type II error}} \geq 1 - \delta.$$

These inequalities give an upper bound on any adversary's accuracy on the task of distinguishing between any two adjacent datasets. We remark that the membership inference problem as we have defined it is a more complicated setting than this hypothesis testing framework. Rather than simply being presented with two possible subsets $D_0$ or $D_1$ and asked to distinguish between only these possibilities, we must distinguish between two large *collections* of datasets: the $\binom{n}{n/2}$ subsets which contain $\mathbf{x}^*$, and the $\binom{n}{n/2}$ subsets which do not. We emphasize this point because it is nontrivial to show that distinguishing between any two datasets at once also allows one to distinguish between two collections of datasets: moving from the simpler "one versus one" to the more complicated "many versus many" setting requires establishing a correspondence between sets in $\mathbb{D}^{\text{in}}$ and sets in $\mathbb{D}^{\text{out}}$. For a complete proof, see Appendix B.

To help interpret this result, we remark that for $\varepsilon \approx 0$, a Taylor expansion of $1/(1 + e^{-\varepsilon})$ about $\varepsilon = 0$ shows that

$$\delta + (1 - \delta)/(1 + e^{-\varepsilon}) - 1/2 \approx \delta/2 + (1 - \delta)\varepsilon/4.$$

Thus in the regime where strong privacy guarantees are required ($\eta \approx 0$), we will need $\delta \approx 0$. Substituting $\delta \approx 0$ into the equation above then yields $\eta \approx \varepsilon/4$.

In fact, DP is *strictly* stronger than MIP, which we make precise with the following theorem.

**Theorem 5.** *For any $\eta > 0$, there exists an algorithm $\mathcal{A}$ which is $\eta$-MIP but not $(\varepsilon, \delta)$-DP for any $\varepsilon < \infty$ and $\delta < 1$.*

In order to better understand the difference between DP and MIP, let us again examine Proposition 2. Recall that this proposition showed that *marginally* over the output of $\mathcal{A}$, the conditional probability that $\mathbf{x}^* \in \mathcal{D}_{\text{train}}$ given the algorithm output should not differ too much from the unconditional probability that $\mathbf{x}^* \in \mathcal{D}_{\text{train}}$. The following proposition shows that (pure) DP requires this condition to hold for *every* output of $\mathcal{A}(\mathcal{D}_{\text{train}})$.

**Proposition 6.** *If $\mathcal{A}$ is an $(\varepsilon, 0)$-DP algorithm, then for any $\mathbf{x}^*$, we have*

$$\frac{\mathbb{P}(\mathbf{x}^* \notin \mathcal{D}_{\text{train}} \mid \mathcal{A}(\mathcal{D}_{\text{train}}))}{\mathbb{P}(\mathbf{x}^* \in \mathcal{D}_{\text{train}} \mid \mathcal{A}(\mathcal{D}_{\text{train}}))} \leq e^{\varepsilon} \frac{\mathbb{P}(\mathbf{x}^* \notin \mathcal{D}_{\text{train}})}{\mathbb{P}(\mathbf{x}^* \in \mathcal{D}_{\text{train}})}.$$

Proposition 6 can be thought of as an extension of the Bayesian interpretation of DP explained by Jordon et al. (2022). Namely, the definition of DP immediately implies that, for any two adjacent sets $D$ and $D'$,

$$\frac{\mathbb{P}(\mathcal{D}_{\text{train}} = D \mid \mathcal{A}(\mathcal{D}_{\text{train}}))}{\mathbb{P}(\mathcal{D}_{\text{train}} = D' \mid \mathcal{A}(\mathcal{D}_{\text{train}}))} \leq e^{\varepsilon} \frac{\mathbb{P}(\mathcal{D}_{\text{train}} = D)}{\mathbb{P}(\mathcal{D}_{\text{train}} = D')}.$$

We remark that the proof of Proposition 6 indicates that converting between the case of distinguishing between two adjacent datasets (as in the inequality above, and as done in (Wasserman and Zhou, 2010;

Kairouz et al., 2015; Nasr et al., 2021)) vs. the case of membership inference is non-trivial: both our proof and a similar one by Thudi et al. (2022) require the construction of a injective function between sets which do/do not contain $\mathbf{x}^*$.

Aside from (approximate) differential privacy, there are other privacy notions such as $f$-DP or Gaussian DP (GDP) (Dong et al., 2019), Renyi DP (RDP) (Mironov, 2017), concentrated DP (CDP) (Dwork and Rothblum, 2016), and truncated CDP (tCDP) (Bun et al., 2018). The following theorem shows that MIP is distinct from each of these notions as well.

**Theorem 7.** *For any $\eta > 0$, there exists an algorithm $\mathcal{A}$ and a dataset $\mathcal{D}$ such that $\mathcal{A}$ is $\eta$-MIP with respect to $\mathcal{D}$, but $\mathcal{A}$ is not (nontrivially) RDP, CDP, tCDP, or $f$-DP.*

**Summary**   The results of this section elucidate the differences and similarities between MIP and DP. For clarity, we summarize them as follows:

1. Both DP and MIP are theoretically principled notions of privacy, with provable limitations on the private information that an adversary can obtain given the output of an algorithm which satisfies either notion.

2. DP implies general bounds on the outcome of *any* privacy attack, while MIP guarantees apply specifically to membership inference attacks and, by extension, any privacy attack which would imply a successful membership inference attack. Thus, MIP guarantees are a *relaxed* form of privacy as compared to DP.

   (a) Theorem 4 shows that DP is at least as strong as MIP by showing that DP implies MIP, and gives the best correspondence possible between the DP and MIP privacy parameters.
   (b) Theorems 5 and 7 show that MIP is strictly weaker than existing forms of DP.

### 3.5   MIP in the Low False Positive Rate Regime

As presented, MIP measures and bounds privacy loss in terms of an adversary's average accuracy advantage over random guess. However, as argued by Carlini et al. (2021a), in some cases it may be more relevant to consider the adversary's true positive rate (TPR) at a low false positive rate (FPR). This corresponds to scenarios in which an adversary cannot identify most individuals but can confidently infer that a small number of individuals belong to the private dataset, which would still constitute a privacy violation even though the average accuracy is low. While MIP is not explicitly defined to address this scenario, MIP actually implies strong results for the low FPR regime as well. This is the content of Theorem 8.

**Theorem 8.** *Suppose that $\mathcal{A}$ is $\eta$-MIP. Then for any adversary $\mathcal{I}$ and any target record $\mathbf{x}^*$, we have that*

$$\mathbb{P}(\mathcal{I}(\mathbf{x}^*, \mathcal{A}(\mathcal{D}_{\text{train}})) = 1 \mid \mathbf{x}^* \in \mathcal{D}_{\text{train}}) \leq \mathbb{P}(\mathcal{I}(\mathbf{x}^*, \mathcal{A}(\mathcal{D}_{\text{train}})) = 1 \mid \mathbf{x}^* \notin \mathcal{D}_{\text{train}}) + \eta.$$

*In other words, any adversary must have* $\text{TPR} \leq \text{FPR} + \eta$.

Theorem 8 shows that MIP guarantees translate into an additive bound on the TPR. Thus, while MIP is defined in terms of average membership inference accuracy, it still provides a meaningful measurement and bound of privacy leakage in the low FPR regime. The connection between the attacker's accuracy and the TPR/FPR gap is not specific to MIP, and indeed the same bound on this gap in terms of the attacker's accuracy was shown by Yeom et al. (2018). However, while the form of our bounds is the same, note that Theorem 8 does not follow from Yeom et al. (2018) because the membership inference setting considered in their work is different from ours. (In particular, they focused on membership inference in the presence of i.i.d. data.)

## 4 Guaranteeing MIP via Noise Addition

In this section, we show that a small modification to standard training procedures can be used to guarantee MIP. Suppose that $\mathcal{A}$ takes as input a data set $D$ and produces output $\theta \in \mathbb{R}^d$. For instance, $\mathcal{A}$ may compute a simple statistical query on $D$, such as mean estimation, but our results apply equally well in the case that e.g. $\mathcal{A}(D)$ are the weights of a neural network trained on $D$. If $\theta$ are the weights of a generative model, then if we can guarantee MIP for $\theta$, then by the data processing inequality (Theorem 3), this guarantees privacy for any output of the generative model.

The distribution over training data (in our case, the uniform distribution over size $n/2$ subsets of our complete dataset $\mathcal{D}$) induces a distribution over the output $\theta$. The question is, what is the smallest amount of noise we can add to $\theta$ which will guarantee MIP? If we add noise on the order of $\max_{D \sim D' \subseteq \mathcal{D}} \|\mathcal{A}(D) - \mathcal{A}(D')\|$, then we can adapt the standard proof for guaranteeing DP in terms of algorithm sensitivity to show that a restricted version of DP (only with respect subsets of $\mathcal{D}$) holds in this case, which in turn guarantees MIP. However, recall that by Propositions 2 and 6, MIP is only asking for a *marginal* guarantee on the change in the posterior probability of $D$ given $A$, whereas DP is asking for a *conditional* guarantee on the posterior. So while max seems necessary for a conditional guarantee, the *moments* of $\theta$ should be sufficient for a marginal guarantee. Theorem 9 shows that this intuition is correct.

**Theorem 9.** *Let $\|\cdot\|$ be any norm, and let $\sigma^M \geq \mathbb{E}\|\theta - \mathbb{E}\theta\|^M$ be an upper bound on the $M$-th central moment of $\theta$ with respect to this norm over the randomness in $\mathcal{D}_{\text{train}}$ and $\mathcal{A}$. Let $X$ be a random variable with density proportional to $\exp(-\frac{1}{c\sigma}\|X\|)$ with $c = (7.5/\eta)^{1+\frac{2}{M}}$. Finally, let $\hat{\theta} = \theta + X$. Then $\hat{\theta}$ is $\eta$-MIP, i.e., for any adversary $\mathcal{I}$,*

$$\mathbb{P}(\mathcal{I}(\mathbf{x}^*, \hat{\theta}) = y^*) \leq 1/2 + \eta.$$

At first glance, Theorem 9 may appear to be adding noise of equal magnitude to all of the coordinates of $\theta$, regardless of how much each contributes to the central moment $\sigma$. However, by carefully selecting the norm $\|\cdot\|$, we can add non-isotropic noise to $\theta$ such that the marginal noise level reflects the variability of each specific coordinate of $\theta$. This is the content of Corollary 10. (GenNormal$(\mu, \alpha, \beta)$ refers to the probability distribution with density proportional to $\exp(-(|x-\mu|/\alpha)^\beta)$.)

**Corollary 10.** *Let $M \geq 2$, $\sigma_i^M \geq \mathbb{E}|\theta_i - \mathbb{E}\theta_i|^M$, and define $\|x\|_{\sigma,M} = \left(\sum_{i=1}^d \frac{|x_i|^M}{d\sigma_i^M}\right)^{1/M}$ for any vector $x = (x_1, \ldots, x_d)^\top$. Generate $Y_i \sim$ GenNormal$(0, \sigma_i, M)$, $U = Y/\|Y\|_{\sigma,M}$ and draw $r \sim$ Laplace$\left((6.16/\eta)^{1+2/M}\right)$. Finally, set $X = rU$ and return $\hat{\theta} = \theta + X$. Then $\hat{\theta}$ is $\eta$-MIP.*

It may be the case that the $\sigma_i$ are not known or an analytic upper bound cannot be easily derived. In such cases, an approximate or asymptotic form of MIP can be employed by estimating the $\sigma_i$ from data. We implement this intuition and devise an approximate method for applying MIP in Algorithm 1. The algorithm has two important caveats. First, we remark briefly that the estimator for $\sigma_j$ used in Algorithm 1 is not unbiased, but it is consistent (i.e., the bias approaches 0 as $B \to \infty$). When $M = 2$, there is a well-known unbiased estimate for the variance which replace $1/B$ with $1/(B-1)$, and one can make similar corrections for general $M$ (Gerlovina and Hubbard, 2019). These corrections generally yield very small differences and the naive estimator presented in the algorithm should suffice. Second, the conditions of Theorem 9 and Corollary 10 require that $\sigma$ or $\sigma_i$ are *upper bounds* on the central moments of the appropriate model parameters. While our estimation procedure for $\sigma$ is consistent and guaranteed to give the correct value as $B \to \infty$ (or if $B = \binom{n}{n/2}$ and all possible subsets $\mathcal{D}_{\text{train}}$ are tested, so that $\sigma$ is computed deterministically and exactly), for finite $B$, it *cannot be guaranteed* that the estimated $\sigma$ is an upper bound on the central moment, therefore the results of Theorem 9 or Corollary 10 cannot be guaranteed to hold. If strict theoretical enforcement of MIP is required, $\sigma$ must be guaranteed to be an upper bound on the central moment, via analytical arguments (as is done in Proposition 11, for instance) or other means. While this is not ideal, we believe it is still an improvement over the sensitivity bound needed for many DP algorithms (e.g., the Laplace mechanism), since $\sigma$ may be consistently estimated from data, but there is no consistent estimator of the sensitivity. The $\sigma$ estimation heuristic can be viewed as similar to the "folklore" statement in the DP literature that $\epsilon \leq 1$ is needed for theory, but $\epsilon \leq 10$ is fine in practice (Ponomareva et al., 2023). It is also conceivable that the effects of a finite $\sigma$ estimation budget $B$ can be taken into account in the resulting MIP parameter $\eta$, but we leave this for future work.

---

**Algorithm 1** MIP via noise addition

---

**Require:** Private dataset $\mathcal{D}$, $\sigma$ estimation budget $B$, MIP parameter $\eta$
  $\mathcal{D}_{\text{train}} \leftarrow \text{RANDOMSPLIT}(\mathcal{D}, 1/2)$

  *# Estimate $\sigma$ if an a priori bound is not known*
  **for** $i = 1, \ldots, B$ **do**
    $\mathcal{D}_{\text{train}}^{(i)} \leftarrow \text{RANDOMSPLIT}(\mathcal{D}_{\text{train}}, 1/2)$
    $\theta^{(i)} \leftarrow \mathcal{A}(\mathcal{D}_{\text{train}}^{(i)})$
  **end for**
  **for** $j = 1, \ldots, d$ **do**
    $\bar{\theta}_j \leftarrow \frac{1}{B} \sum_{i=1}^{B} \theta_j^{(i)}$
    $\sigma_j \leftarrow \left( \frac{1}{B} \sum_{i=1}^{B} (\theta_j^{(i)} - \bar{\theta}_j)^M \right)^{1/M}$
  **end for**

  *# Add appropriate noise to the base algorithm's output*
  $U \leftarrow \text{Unif}(\{u \in \mathbb{R}^d \ : \ \|u\|_{\sigma,M} = 1\})$
  $r \leftarrow \text{Laplace}\left( \left( \frac{6.16}{\eta} \right)^{1+2/M} \right)$
  $X \leftarrow rU$

  **return** $\mathcal{A}(\mathcal{D}_{\text{train}}) + X$

---

**When Does MIP Require Less Noise Than DP?** By Theorem 4, any DP algorithm gives rise to a MIP algorithm, so we *never* need to add more noise than the amount required to guarantee DP, in order to guarantee MIP. However, Theorem 9 shows that MIP affords an advantage over DP when the variance of our algorithm's output (over subsets of size $n/2$) is much smaller than its sensitivity $\Delta$, which is defined as the maximum change in the algorithm's output when evaluated on two datasets which differ in only one element. For instance, applying the Laplace mechanism from DP requires a noise which scales like $\Delta/\epsilon$ to guarantee $\varepsilon$-DP. It is easy to construct examples where the variance is much smaller than the sensitivity if the output of our "algorithm" is allowed to be completely arbitrary as a function of the input. However, it is more interesting to ask if there are any *natural* settings in which this occurs. Proposition 11 answers this question in the affirmative by demonstrating an estimation procedure where MIP can require arbitrarily smaller noise than that required by DP.

**Proposition 11.** *For any finite $D \subseteq \mathbb{R}$, define $\mathcal{A}(D) = \frac{1}{\sum_{x \in D} x}$. Given a dataset $\mathcal{D}$ of size $n$, define* $\mathbb{D} = \{D \subseteq \mathcal{D} \ : \ |D| = \lfloor n/2 \rfloor\}$*, and define*

$$\sigma^2 = \text{Var}(\mathcal{A}(D)), \qquad \Delta = \max_{D \sim D' \in \mathbb{D}} |\mathcal{A}(D) - \mathcal{A}(D')|.$$

*Here the variance is taken over $D \sim \text{Unif}(\mathbb{D})$. Then for all $n$, there exists a dataset $\mathcal{D}$ with $|\mathcal{D}| = n$ such that $\sigma^2 = O(1)$ but $\Delta = \Omega(2^{n/3})$.*

In Appendix D, we show that the dataset

$$\mathcal{D} = \{2^i \ : \ i = 0, \ldots, n-2\} \cup \left\{ \binom{n}{n/2}^{1/2} - \sum_{i=0}^{\frac{n}{2}-2} 2^i \right\}$$

satisfies the conditions of Proposition 11. To illustrate this result, we plot the noise level needed to guarantee MIP vs. the corresponding level of DP (with the correspondence given by Theorem 4) for this construction in Fig. 1

Refer to Fig. 1. Dotted lines refer to DP, while the solid line is for MIP with $M = 2$. The $x$-axis gives the best possible bound on the attacker's improvement in accuracy over random guessing–i.e., the parameter $\eta$

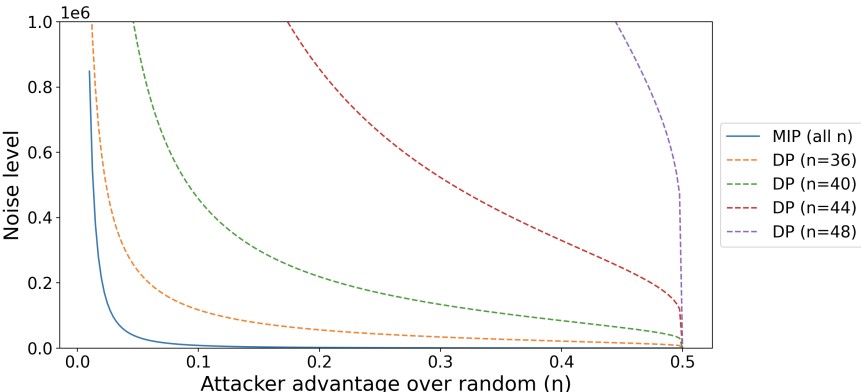

Figure 1: Noise level vs. privacy guarantee for MIP and DP for the construction used to prove Proposition 11 (lower is better). For datasets with at least $n = 36$ points and for almost all values of $\eta$, MIP allows us to add much less noise than what would be required by naively applying DP, i.e., by first enforcing $\varepsilon$-DP by bounding the algorithm sensitivity and applying the Laplace mechanism, then using the tight DP/MIP correspondence from Theorem 4. For $n > 48$, the amount of noise required by DP is so large that it will not appear on the plot.

for an $\eta$-MIP method–according to that method's guarantees. For DP, the value along the $x$-axis is given by the (tight) correspondence in Theorem 4, namely $\eta = \frac{1}{1+e^{-\varepsilon}} - \frac{1}{2}$. $\eta = 0$ corresponds to perfect privacy (the attacker cannot do any better than random guessing), while $\eta = \frac{1}{2}$ corresponds to no privacy (the attacker can determine membership with perfect accuracy). The $y$-axis denotes the amount of noise that must be added to the non-private algorithm's output, as measured by the scale parameter of the Laplace noise that must be added (lower is better). For MIP, by Theorem 9, this is $(6.16/\eta)^2\sigma$ where $\sigma$ is an upper bound on the variance of the base algorithm over random subsets, and for DP this is $\frac{\Delta}{\log\frac{1+2\eta}{1-2\eta}}$. (This comes from solving $\eta = \frac{1}{1+e^{-\varepsilon}} - \frac{1}{2}$ for $\varepsilon$, then using the fact that Laplace($\Delta/\varepsilon$) noise must be added to guarantee $\varepsilon$-DP.) For DP, the amount of noise necessary changes with the size $n$ of the private dataset. For MIP, the amount of noise does not change, so there is only one line.

The results show that for even small datasets ($n \geq 36$) and for $\eta \geq 0.01$, direct noise accounting for MIP gives a large advantage over guaranteeing MIP via DP. In practice, such small datasets are uncommon. For $n$ in the range typical for a ML dataset (10,000+), the noise required for DP is many orders of magnitude larger than that required for MIP, and is not even visible on the plot. (Refer to Proposition 11. The noise required for DP grows exponentially in $n$, while it remains constant in $n$ for MIP.)

## 5 Conclusion

In this work, we proposed a novel privacy property, membership inference privacy (MIP) and explained its properties and relationship with differential privacy (DP). The MIP property is more readily interpretable than the guarantees offered by (DP). In some cases, MIP also requires a smaller amount of noise to guarantee as compared to DP, allowing for greater utility on the base task. We proposed a simple "wrapper" method for guaranteeing MIP, which can be implemented with a minor modification both to simple statistical queries or more complicated tasks such as the training procedure for parametric machine learning models. In many "low stakes" scenarios where privacy is nevertheless desired, weaker privacy notions such as MIP may be appropriate. For instance, in recommender systems or online advertising placement, user data is less sensitive than patient data in a healthcare setting or client data in finance. The decision of what level of privacy to enforce should be made with the help of domain experts on a case-by-case basis, and MIP expands the range of available options.

**Limitations** As the example used to prove Theorem 5 shows, there are cases where algorithms may have certain (low probability) non-private outputs, but which still satisfy MIP. Thus, algorithms which satisfy MIP may require post-processing to ensure that the output is not one of the low-probability events in which data privacy is leaked. In addition, because MIP is determined with respect to a holdout set still drawn from $\mathcal{D}$, an adversary may be able to determine with high probability whether a given sample was contained in $\mathcal{D}$, rather than just in $\mathcal{D}_{\text{train}}$, if $\mathcal{D}$ is sufficiently different from the rest of the population.

Another limitation of MIP is that it does not necessarily protect against all possible privacy attacks. MIP protects against any privacy attack which would *imply* a successful membership inference attack. One instance of this is reconstruction attacks (discussed in the Section 2), where the adversary seeks to reconstruct training data "ex nihilo" rather than being *supplied* with potentially included data. If an attacker can successfully reconstruct the training data, then a simple membership inference attack is to reconstruct the training data first and check whether or not the attacked record is contained in the reconstruction. If membership inference is impossible by enforcing MIP, then this attack cannot succeed with high probability, so reconstruction cannot succeed with high probability either. On the other hand, for some versions of linkage attacks, MIP may not provide protection. For instance, in a standard version of a linkage attack, the adversary already knows that a specific individual was contained in the training dataset, just not which specific *record* belongs to that individual. Such an attacker has already passed the "line of defense" enforced by MIP (namely, the membership of an individual in the training data), so MIP will not apply to this scenario.

**Future Work** Theorem 4 suggests that DP implies MIP in general. However, Theorem 9 shows that a finer-grained analysis of a standard DP mechanism (the Laplace mechanism) is possible, showing that we can guarantee MIP with less noise. It seems plausible that a similar analysis can be undertaken for other DP mechanisms. In addition to these "wrapper" type methods which can be applied on top of existing algorithms, bespoke algorithms for guaranteeing MIP in particular applications (such as synthetic data generation) are also of interest. Noise addition is a simple and effective way to enforce privacy, but other classes of mechanisms may also be possible. For instance, it would be interesting to study whether or not it is possible to directly regularize a probabilistic model using Proposition 2.

In Theorem 3, we showed that MIP obeys a post-processing inequality, a feature shared by DP. Besides post-processing, DP also has well-studied group privacy and composition properties, and DP guarantees can be amplified by subsampling. An important direction for future work is to establish to what extent similar properties exist for MIP.

Lastly, this paper focused on developing on the theoretical principles and guarantees of MIP. Taking advantage of the relaxed requirements of MIP to develop practical algorithms, and systematic empirical evaluation of these algorithms, is an important direction for future work.

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

## A Proofs for Section 3.3

**Proposition 2.** *Let* $\mathbb{A} = \text{Range}(\mathcal{A})$ *and for simplicity assume that* $\mathbb{A}$ *is discrete. Let* $\mathbb{P}(\mathcal{A}(\mathcal{D}_{\text{train}}) = A)$ *denote the probability of a given output* $A \in \mathbb{A}$ *over the randomness in* $\mathcal{D}_{\text{train}}$ *and* $\mathcal{A}$*. Then* $\mathcal{A}$ *is* $\eta$*-MIP if and only if*

$$\sum_{A \in \mathbb{A}} \Big( \max \big\{ \mathbb{P}(\mathbf{x}^* \in \mathcal{D}_{\text{train}} \,|\, \mathcal{A}(\mathcal{D}_{\text{train}}) = A), \mathbb{P}(\mathbf{x}^* \notin \mathcal{D}_{\text{train}} \,|\, \mathcal{A}(\mathcal{D}_{\text{train}}) = A) \big\} \, \mathbb{P}(\mathcal{A}(\mathcal{D}_{\text{train}}) = A) \Big) \leq \frac{1}{2} + \eta.$$

*Furthermore, the optimal adversary is given by*

$$\mathcal{I}(\mathbf{x}^*, A) = \mathbb{1}\{\mathbb{P}(\mathbf{x}^* \in \mathcal{D}_{\text{train}} \,|\, \mathcal{A}(\mathcal{D}_{\text{train}}) = A) \geq 1/2\}.$$

*Proof.* To avoid measure theoretic complications, we will show the result for the case where the output space $\mathbb{A}$ of $\mathcal{A}$ is discrete. The results extend in a straightforward way to continuous output, with probability masses replaced with densities and sums replaced with integrals.

We will show that the membership inference attack specified in the theorem is optimal, then compute the resulting probability of membership inference. Let $C = \mathbb{P}(\mathcal{D}_{\text{train}} = D) = \binom{n}{k}^{-1}$ be the constant probability that $\mathcal{D}_{\text{train}}$ is equal to any particular training set $D \in \mathbb{D}$. We have

$$\mathbb{P}(\mathcal{I}(\mathbf{x}^*, \mathcal{A}(\mathcal{D}_{\text{train}})) = y^*) = \sum_{D \in \mathbb{D}} \mathbb{P}(\mathcal{D}_{\text{train}} = D) \sum_{A \in \mathbb{A}} \mathbb{P}(\mathcal{A}(D) = A) \cdot \mathbb{P}(\mathcal{I}(\mathbf{x}^*, A) = \mathbb{1}\{\mathbf{x}^* \in D\})$$

$$= C \sum_{A \in \mathbb{A}} \left[ \sum_{D \in \mathbb{D}^{\text{in}}} \mathbb{P}(\mathcal{A}(D) = A) \cdot \mathbb{P}(\mathcal{I}(\mathbf{x}^*, A) = 1) + \sum_{D \in \mathbb{D}^{\text{out}}} \mathbb{P}(\mathcal{A}(D) = A) \cdot (1 - \mathbb{P}(\mathcal{I}(\mathbf{x}^*, A) = 1)) \right]$$

$$= C \sum_{A \in \mathbb{A}} \left[ \left( \sum_{D \in \mathbb{D}^{\text{in}}} \mathbb{P}(\mathcal{A}(D) = A) - \sum_{D \in \mathbb{D}^{\text{out}}} \mathbb{P}(\mathcal{A}(D) = A) \right) \mathbb{P}(\mathcal{I}(\mathbf{x}^*, A) = 1) + \sum_{D \in \mathbb{D}^{\text{out}}} \mathbb{P}(\mathcal{A}(D) = A) \right]. \quad (2)$$

The choice of algorithm $\mathcal{I}$ just specifies the value of $\mathbb{P}(\mathcal{I}(\mathbf{x}^*, A) = 1)$ for each sample $\mathbf{x}^*$ and each $A \in \mathbb{A}$. We see that the maximum membership inference probability is obtained when

$$\mathbb{P}(\mathcal{I}(\mathbf{x}^*, A) = 1) = \mathbb{1} \left\{ \sum_{D \in \mathbb{D}^{\text{in}}} \mathbb{P}(\mathcal{A}(D) = A) - \sum_{D \in \mathbb{D}^{\text{out}}} \mathbb{P}(\mathcal{A}(D) = A) \geq 0 \right\}. \quad (3)$$

Observe that

$$\mathbb{P}(\mathbf{x}^* \in \mathcal{D}_{\text{train}} \mid \mathcal{A}(\mathcal{D}_{\text{train}}) = A)$$

$$= \frac{\sum_{D \in \mathbb{D}^{\text{in}}} \mathbb{P}(\mathcal{D}_{\text{train}} = D) \cdot \mathbb{P}(\mathcal{A}(D) = A)}{\sum_{D \in \mathbb{D}^{\text{in}}} \mathbb{P}(\mathcal{D}_{\text{train}} = D) \cdot \mathbb{P}(\mathcal{A}(D) = A) + \sum_{D \in \mathbb{D}^{\text{out}}} \mathbb{P}(\mathcal{D}_{\text{train}} = D) \cdot \mathbb{P}(\mathcal{A}(D) = A)}$$

$$= \frac{\sum_{D \in \mathbb{D}^{\text{in}}} \mathbb{P}(\mathcal{A}(D) = A)}{\sum_{D \in \mathbb{D}^{\text{in}}} \mathbb{P}(\mathcal{A}(D) = A) + \sum_{D \in \mathbb{D}^{\text{out}}} \mathbb{P}(\mathcal{A}(D) = A)}. \quad (4)$$

We note that (4) is at least $1/2$ if and only if $\sum_{D \in \mathbb{D}^{\text{in}}} \mathbb{P}(\mathcal{A}(D) = A) \geq \sum_{D \in \mathbb{D}^{\text{out}}} \mathbb{P}(\mathcal{A}(D) = A)$, establishing the equivalence between (3) and the stated optimal adversary.

Next, we use the optimal adversary from (3) to upper bound the accuracy of any adversary. Plugging the optimal adversary into (2) and recalling that $C = \mathbb{P}(\mathcal{D}_{\text{train}} = D)$, we have

$$\mathbb{P}(\mathcal{I}(\mathbf{x}^*, \mathcal{A}(\mathcal{D}_{\text{train}})) = y^*)$$

$$\leq \sum_{A \in \mathbb{A}} \max \left\{ \sum_{D \in \mathbb{D}^{\text{in}}} \mathbb{P}(\mathcal{D}_{\text{train}} = D) \mathbb{P}(\mathcal{A}(D) = A), \sum_{D \in \mathbb{D}^{\text{out}}} \mathbb{P}(\mathcal{D}_{\text{train}} = D) \mathbb{P}(\mathcal{A}(D) = A) \right\}$$

$$= \sum_{A \in \mathbb{A}} \max \left\{ \mathbb{P}(\mathbf{x}^* \in \mathcal{D}_{\text{train}} \wedge \mathcal{A}(\mathcal{D}_{\text{train}}) = A), \mathbb{P}(\mathbf{x}^* \notin \mathcal{D}_{\text{train}} \wedge \mathcal{A}(\mathcal{D}_{\text{train}}) = A) \right\}$$

$$= \sum_{A \in \mathbb{A}} \max \left\{ \frac{\mathbb{P}(\mathbf{x}^* \in \mathcal{D}_{\text{train}} \wedge \mathcal{A}(\mathcal{D}_{\text{train}}) = A)}{\mathbb{P}(\mathcal{A}(\mathcal{D}_{\text{train}}) = A)}, \frac{\mathbb{P}(\mathbf{x}^* \notin \mathcal{D}_{\text{train}} \wedge \mathcal{A}(\mathcal{D}_{\text{train}}) = A)}{\mathbb{P}(\mathcal{A}(\mathcal{D}_{\text{train}}) = A)} \right\} \mathbb{P}(\mathcal{A}(\mathcal{D}_{\text{train}}) = A)$$

$$= \sum_{A \in \mathbb{A}} \max \left\{ \mathbb{P}(\mathbf{x}^* \in \mathcal{D}_{\text{train}} \mid \mathcal{A}(\mathcal{D}_{\text{train}}) = A), \mathbb{P}(\mathbf{x}^* \notin \mathcal{D}_{\text{train}} \mid \mathcal{A}(\mathcal{D}_{\text{train}}) = A) \right\} \mathbb{P}(\mathcal{A}(\mathcal{D}_{\text{train}}) = A). \quad (5)$$

The expression in (5) is precisely the formulation of MIP given in Proposition 2. This completes the proof. $\square$

# B   Proofs for Section 3.4

In what follows, we will assume without loss of generality that $k \geq n/2$. The proofs in the case $k < n/2$ are almost identical and can be obtained by simply swapping $\mathbb{D}^{\text{in}} \leftrightarrow \mathbb{D}^{\text{out}}$ and $k \leftrightarrow n - k$.

**Lemma 12.** *Fix $\mathbf{x}^* \in \mathcal{D}$ and let $\mathbb{D}^{\text{in}} = \{D \in \mathcal{D} \mid \mathbf{x}^* \in D\}$ and $\mathbb{D}^{\text{out}} = \{D \in \mathcal{D} \mid \mathbf{x}^* \notin D\}$. If $k \geq n/2$ then there is an injective function $f : \mathbb{D}^{\text{out}} \to \mathbb{D}^{\text{in}}$ such that $D \sim f(D)$ for all $D \in \mathbb{D}^{\text{out}}$.*

*Proof.* We define a bipartite graph $G$ on nodes $\mathbb{D}^{\text{in}}$ and $\mathbb{D}^{\text{out}}$. There is an edge between $D^{\text{in}} \in \mathbb{D}^{\text{in}}$ and $D^{\text{out}} \in \mathbb{D}^{\text{out}}$ if $D^{\text{out}}$ can be obtained from $D^{\text{in}}$ by removing $\mathbf{x}^*$ from $D^{\text{in}}$ and replacing it with another element, i.e. if $D^{\text{in}} \sim D^{\text{out}}$. To prove the lemma, it suffices to show that there is a matching on $G$ which covers $D^{\text{out}}$. We will show this via Hall's marriage theorem.

First, observe that $G$ is a $(k, n-k)$-biregular graph. Each $D^{\text{in}} \in \mathbb{D}^{\text{in}}$ has $n-k$ neighbors which are obtained from $D^{\text{in}}$ by selecting which of the remaining $n-k$ elements to replace $\mathbf{x}^*$ with; each $D^{\text{out}} \in \mathbb{D}^{\text{out}}$ has $k$ neighbors which are obtained by selecting which of the $k$ elements in $D^{\text{out}}$ to replace with $\mathbf{x}^*$.

Let $W \subseteq \mathbb{D}^{\text{out}}$ and let $N(W) \subseteq \mathbb{D}^{\text{in}}$ denote the neighborhood of $W$. We have the following:

$$|N(W)| = \sum_{D^{\text{in}} \in N(W)} \frac{\sum_{D^{\text{out}} \in W} \mathbb{1}\{D^{\text{out}} \sim D^{\text{in}}\}}{\sum_{D^{\text{out}} \in W} \mathbb{1}\{D^{\text{out}} \sim D^{\text{in}}\}}$$

$$\geq \sum_{D^{\text{in}} \in N(W)} \frac{\sum_{D^{\text{out}} \in W} \mathbb{1}\{D^{\text{out}} \sim D^{\text{in}}\}}{\sum_{D^{\text{out}} \in \mathbb{D}^{\text{out}}} \mathbb{1}\{D^{\text{out}} \sim D^{\text{in}}\}}$$

$$= \frac{1}{n-k} \sum_{D^{\text{out}} \in W} \sum_{D^{\text{in}} \in N(W)} \mathbb{1}\{D^{\text{out}} \sim D^{\text{in}}\} \tag{6}$$

$$= \frac{k}{n-k} |W|. \tag{7}$$

Equation (6) holds since each $D^{\text{in}}$ has degree $n-k$ and by exchanging the order of summation. Similarly, (7) holds since each $D^{\text{out}}$ has degree $k$. When $k \geq n/2$, we thus have $|N(W)| \geq |W|$ for every $W \subseteq \mathbb{D}^{\text{out}}$ and the result follows by Hall's marriage theorem. $\qquad\square$

**Theorem 4.** *Let $\mathcal{A}$ be $(\varepsilon, \delta)$-DP. Then $\mathcal{A}$ is $\eta$-MIP with $\eta = \delta + \frac{1-\delta}{1+e^{-\varepsilon}} - \frac{1}{2}$. Furthermore, this bound is tight, i.e. for any $\varepsilon > 0$ and $0 \leq \delta < 1$, there exists an $(\varepsilon, \delta)$-DP algorithm against which the optimal attacker has accuracy $\delta + \frac{1-\delta}{1+e^{-\varepsilon}}$.*

*Proof.* Note that any identification algorithm $\mathcal{I}(\mathbf{x}^*, \mathcal{A}(\mathcal{D}_{\text{train}}))$ is specified by a "rejection region" $S$ for each $\mathbf{x}^*$. That is, with $\mathbf{x}^*$ fixed, there is a subset $S$ of the range of $\mathcal{A}$ such that $\mathcal{I}$ predicts $\mathbf{x}^* \notin \mathcal{D}_{\text{train}}$ iff $\mathcal{A}(\mathcal{D}_{\text{train}}) \in S$.

Let $D_0 \sim D_1$ with $\mathbf{x}^* \in D_0$ and $\mathbf{x}^* \notin D_1$. By Theorem 2.1 of Kairouz et al. (2015), we have that

$$\mathbb{P}(\mathcal{A}(D_0) \in S) + e^{\varepsilon} \mathbb{P}(\mathcal{A}(D_1) \notin S) \geq 1 - \delta,$$

$$e^{\varepsilon} \mathbb{P}(\mathcal{A}(D_0) \in S) + \mathbb{P}(\mathcal{A}(D_1) \notin S) \geq 1 - \delta.$$

Summing these inequalities, we have that

$$\mathbb{P}(\mathcal{A}(D_0) \in S) + \mathbb{P}(\mathcal{A}(D_1) \notin S) \geq \frac{2(1-\delta)}{1+e^{\varepsilon}}. \tag{8}$$

We can now use this to lower bound the probability that the adversary is incorrect:

$$\mathbb{P}(\text{adversary is incorrect}) = \sum_{D_0 \in \mathbb{D}^{\text{in}}} \mathbb{P}(\mathcal{D}_{\text{train}} = D_0)\mathbb{P}(\mathcal{A}(D_0) \in S)$$

$$+ \sum_{D_1 \in \mathbb{D}^{\text{out}}} \mathbb{P}(\mathcal{D}_{\text{train}} = D_1)\mathbb{P}(\mathcal{A}(D_1) \notin S)$$

$$= \sum_{D_0 \in \mathbb{D}^{\text{in}}} \binom{n}{n/2}^{-1} (\mathbb{P}(\mathcal{A}(D_0) \in S) + \mathbb{P}(\mathcal{A}(f(D_0)) \notin S)) \tag{9}$$

$$\geq \frac{1}{2}\binom{n}{n/2} \cdot \binom{n}{n/2}^{-1} \frac{2(1-\delta)}{1+e^\varepsilon} = \frac{1-\delta}{1+e^\varepsilon}. \tag{10}$$

Here inequality (9) critically uses the fact that $f$ is a bijection. Since $D_0 \sim f(D_0)$, we can apply inequality (8) to obtain (10). Thus, the adversary's accuracy is at most

$$1 - \frac{1-\delta}{1+e^\varepsilon} = \delta + \frac{1-\delta}{1+e^{-\varepsilon}}$$

as desired.

To see that the bound is tight, consider the following scenario. We take $\mathcal{D} = \{0, 1\}$ and define

$$\mathcal{A}(\{0\}) = \begin{cases} 0 & \text{w.p.} & \delta \\ * & \text{w.p.} & \frac{1-\delta}{1+e^{-\varepsilon}} \\ \dagger & \text{w.p.} & \frac{1-\delta}{1+e^\varepsilon} \end{cases}, \qquad \mathcal{A}(\{1\}) = \begin{cases} 1 & \text{w.p.} & \delta \\ * & \text{w.p.} & \frac{1-\delta}{1+e^\varepsilon} \\ \dagger & \text{w.p.} & \frac{1-\delta}{1+e^{-\varepsilon}} \end{cases}.$$

Note that the algorithm $\mathcal{A}$ is $(\varepsilon, \delta)$-DP. Suppose WLOG that $\mathbf{x}^* = 0$. By the formula from Proposition 2, when $\mathcal{D}_{\text{train}} = \{0\}$, the optimal adversary correctly predicts $\mathbf{x}^* \in \mathcal{D}_{\text{train}}$ when $\mathcal{A}(\{0\}) = 0$ or $*$. When $\mathcal{D}_{\text{train}} = \{1\}$, the optimal adversary correctly predicts $\mathbf{x}^* \notin \mathcal{D}_{\text{train}}$ when $\mathcal{A}(\{1\}) = 1$ or $\dagger$. This gives total accuracy

$$\underbrace{\frac{1}{2}}_{\mathcal{D}_{\text{train}}=\{0\}} \left( \underbrace{\delta}_{\mathcal{A}(\{0\})=0} + \underbrace{\frac{1-\delta}{1+e^{-\varepsilon}}}_{\mathcal{A}(\{0\})=*} \right) + \underbrace{\frac{1}{2}}_{\mathcal{D}_{\text{train}}=\{1\}} \left( \underbrace{\delta}_{\mathcal{A}(\{1\})=1} + \underbrace{\frac{1-\delta}{1+e^{-\varepsilon}}}_{\mathcal{A}(\{1\})=\dagger} \right) = \delta + \frac{1-\delta}{1+e^{-e}},$$

which matches the upper bound in the theorem. $\square$

**Theorem 5.** *For any $\eta > 0$, there exists an algorithm $\mathcal{A}$ which is $\eta$-MIP but not $(\varepsilon, \delta)$-DP for any $\varepsilon < \infty$ and $\delta < 1$.*

*Proof sketch.* Let $n$ be an even, positive integer, let $\mathcal{D} = [n] = \{1, 2, \ldots, n\}$, and set $D^* = [n/2]$. Define $\mathcal{A}(\mathcal{D}_{\text{train}}) = \mathbb{1}\{\mathcal{D}_{\text{train}} = D^*\}$. Clearly, $\mathcal{A}$ is not $(\varepsilon, \delta)$-DP for any $\delta < 1$ (it is a deterministic algorithm with non-constant output). However, the output of $\mathcal{A}$ only lets the adversary determine whether $\mathcal{D}_{\text{train}} = D^*$. As $n$ gets large, marginally over $\mathcal{D}_{\text{train}}$, the advantage over random guessing goes to 0. $\square$

**Proposition 6.** *If $\mathcal{A}$ is an $(\varepsilon, 0)$-DP algorithm, then for any $\mathbf{x}^*$, we have*

$$\frac{\mathbb{P}(\mathbf{x}^* \notin \mathcal{D}_{\text{train}} \mid \mathcal{A}(\mathcal{D}_{\text{train}}))}{\mathbb{P}(\mathbf{x}^* \in \mathcal{D}_{\text{train}} \mid \mathcal{A}(\mathcal{D}_{\text{train}}))} \leq e^\varepsilon \frac{\mathbb{P}(\mathbf{x}^* \notin \mathcal{D}_{\text{train}})}{\mathbb{P}(\mathbf{x}^* \in \mathcal{D}_{\text{train}})}.$$

*Proof.* We have

$$\frac{\mathbb{P}(\mathbf{x}^* \notin \mathcal{D}_{\text{train}} \mid \mathcal{A}(\mathcal{D}_{\text{train}}) = A)}{\mathbb{P}(\mathbf{x}^* \in \mathcal{D}_{\text{train}} \mid \mathcal{A}(\mathcal{D}_{\text{train}}) = A)} = \frac{\sum_{D \in \mathbb{D}^{\text{out}}} \mathbb{P}(\mathcal{A}(D) = A)}{\sum_{D \in \mathbb{D}^{\text{in}}} \mathbb{P}(\mathcal{A}(D) = A)}$$

$$\leq \frac{e^\varepsilon \sum_{D \in \mathbb{D}^{\text{out}}} \min_{D' \in \mathbb{D}^{\text{in}}, D' \sim D} \mathbb{P}(\mathcal{A}(D') = A)}{\sum_{D \in \mathbb{D}^{\text{in}}} \mathbb{P}(\mathcal{A}(D) = A)}.$$

We now analyze this latter expression. We refer again to the biregular graph $G$ defined in Lemma 12. For $D \in \mathbb{D}^{\text{out}}$, $N(D) \subseteq \mathbb{D}^{\text{in}}$ refers to the neighbors of $D$ in $G$, and recall that $|N(D)| = k$ for all $D \in \mathbb{D}^{\text{out}}$. Note that since each $D' \in \mathbb{D}^{\text{in}}$ has $n - k$ neighbors, we have

$$\sum_{D \in \mathbb{D}^{\text{out}}} \sum_{D' \in N(D)} \mathbb{P}(\mathcal{A}(D') = A) = (n - k) \sum_{D' \in \mathbb{D}^{\text{in}}} \mathbb{P}(\mathcal{A}(D') = A).$$

Using this equality, we have

$$\frac{\sum_{D \in \mathbb{D}^{\text{out}}} \min_{D' \in \mathbb{D}^{\text{in}}, D' \sim D} \mathbb{P}(\mathcal{A}(D') = A)}{\sum_{D \in \mathbb{D}^{\text{in}}} \mathbb{P}(\mathcal{A}(D) = A)} = \frac{\sum_{D \in \mathbb{D}^{\text{out}}} \min_{D' \in N(D)} \mathbb{P}(\mathcal{A}(D') = A)}{\frac{1}{n - k} \sum_{D \in \mathbb{D}^{\text{out}}} \underbrace{\sum_{D' \in N(D)} \mathbb{P}(\mathcal{A}(D') = A)}_{\geq k \min_{D' \in N(D)} \mathbb{P}(\mathcal{A}(D') = A)}}$$

$$\leq \frac{n - k}{k}.$$

Since $\mathbb{P}(\mathbf{x}^* \notin \mathcal{D}_{\text{train}}) = \binom{n-1}{k} / \binom{n}{k}$ and $\mathbb{P}(\mathbf{x}^* \in \mathcal{D}_{\text{train}}) = \binom{n-1}{k-1} / \binom{n}{k}$, we have $\frac{\mathbb{P}(\mathbf{x}^* \notin \mathcal{D}_{\text{train}})}{\mathbb{P}(\mathbf{x}^* \in \mathcal{D}_{\text{train}})} = \frac{n-k}{k}$. This completes the proof. $\qquad \square$

**Theorem 7.** *For any $\eta > 0$, there exists an algorithm $\mathcal{A}$ and a dataset $\mathcal{D}$ such that $\mathcal{A}$ is $\eta$-MIP with respect to $\mathcal{D}$, but $\mathcal{A}$ is not (nontrivially) RDP, CDP, tCDP, or $f$-DP.*

*Proof sketch.* The counterexample provided in the proof of Theorem 5 also works for each of these other notions of privacy. $\qquad \square$

## C  Proofs for Section 3.5

**Theorem 8.** *Suppose that $\mathcal{A}$ is $\eta$-MIP. Then for any adversary $\mathcal{I}$ and any target record $\mathbf{x}^*$, we have that*

$$\mathbb{P}(\mathcal{I}(\mathbf{x}^*, \mathcal{A}(\mathcal{D}_{\text{train}})) = 1 \mid \mathbf{x}^* \in \mathcal{D}_{\text{train}}) \leq \mathbb{P}(\mathcal{I}(\mathbf{x}^*, \mathcal{A}(\mathcal{D}_{\text{train}})) = 1 \mid \mathbf{x}^* \notin \mathcal{D}_{\text{train}}) + \eta.$$

*In other words, any adversary must have* TPR$\leq$FPR $+ \eta$.

*Proof.* We will prove the contrapositive statement. Suppose that $\mathcal{I}$ is an adversary that achieves TPR $>$ FPR $+ \eta$ against some algorithm $\mathcal{A}$. By definition, this means that

$$\sum_{D \in \mathbb{D}^{\text{in}}} \mathbb{P}(\mathcal{D}_{\text{train}} = D) \cdot \mathbb{P}(\mathcal{I}(\mathbf{x}^*, \mathcal{A}(D)) = 1) > \sum_{D \in \mathbb{D}^{\text{out}}} \mathbb{P}(\mathcal{D}_{\text{train}} = D) \cdot \mathbb{P}(\mathcal{I}(\mathbf{x}^*, \mathcal{A}(D)) = 1) + \eta$$

$$= \sum_{D \in \mathbb{D}^{\text{out}}} \mathbb{P}(\mathcal{D}_{\text{train}} = D) \cdot (1 - \mathbb{P}(\mathcal{I}(\mathbf{x}^*, \mathcal{A}(D)) = 0)) + \eta$$

$$= \frac{1}{2} - \sum_{D \in \mathbb{D}^{\text{out}}} \mathbb{P}(\mathcal{D}_{\text{train}} = D) \cdot \mathbb{P}(\mathcal{I}(\mathbf{x}^*, \mathcal{A}(D)) = 0) + \eta.$$

Rearranging this inequality yields that

$$\sum_{D \in \mathbb{D}^{\text{in}}} \mathbb{P}(\mathcal{D}_{\text{train}} = D) \cdot \mathbb{P}(\mathcal{I}(\mathbf{x}^*, \mathcal{A}(D)) = 1) + \sum_{D \in \mathbb{D}^{\text{out}}} \mathbb{P}(\mathcal{D}_{\text{train}} = D) \cdot \mathbb{P}(\mathcal{I}(\mathbf{x}^*, \mathcal{A}(D)) = 0) > \frac{1}{2} + \eta,$$

which is precisely the statement that $\mathbb{P}(\mathcal{I}(\mathbf{x}^*, \mathcal{A}(\mathcal{D}_{\text{train}})) = y^*) > 1/2 + \eta$, i.e., that $\mathcal{A}$ is not $\eta$-MIP. $\qquad \square$

## D  Proofs for Section 4

In the proof of Theorem 9, we make use of a generalized Chebyshev inequality.

**Lemma 13** (Chebyshev's Inequality)**.** *Let $\|\cdot\|$ be any norm and $X$ be a random vector with $\mathbb{E}\|X - \mathbb{E}X\|^2 \leq \sigma^k$. Then for any $t > 0$, we have*

$$\mathbb{P}(\|X - \mathbb{E}X\| > t\sigma) \leq 1/t^k.$$

*Proof.* This follows almost directly from Markov's inequality:

$$\mathbb{P}(\|X - \mathbb{E}X\| > t\sigma) = \mathbb{P}(\|X - \mathbb{E}X\|^k > t^k\sigma^k) \leq \frac{\mathbb{E}\|X - \mathbb{E}X\|^k}{t^k\sigma^k} \leq 1/t^k.$$

$\square$

**Theorem 9.** *Let $\|\cdot\|$ be any norm, and let $\sigma^M \geq \mathbb{E}\|\theta - \mathbb{E}\theta\|^M$ be an upper bound on the $M$-th central moment of $\theta$ with respect to this norm over the randomness in $\mathcal{D}_{\text{train}}$ and $\mathcal{A}$. Let $X$ be a random variable with density proportional to $\exp(-\frac{1}{c\sigma}\|X\|)$ with $c = (7.5/\eta)^{1+\frac{2}{M}}$. Finally, let $\hat{\theta} = \theta + X$. Then $\hat{\theta}$ is $\eta$-MIP, i.e., for any adversary $\mathcal{I}$,*

$$\mathbb{P}(\mathcal{I}(\mathbf{x}^*, \hat{\theta}) = y^*) \leq 1/2 + \eta.$$

Before proceeding to the full proof, we provide a sketch.

*Proof sketch.* The proof proceeds by bounding the posterior likelihood ratio $\frac{\mathbb{P}(\mathbf{x}^* \notin \mathcal{D}_{\text{train}} \mid \hat{\theta})}{\mathbb{P}(\mathbf{x}^* \in \mathcal{D}_{\text{train}} \mid \hat{\theta})}$ from above and below for all $\hat{\theta}$ in a large $\|\cdot\|$-ball. This in turn yields an upper bound on the max in the integrand in Proposition 2 with high probability over $\mathcal{A}(\mathcal{D}_{\text{train}})$. The central moment $\sigma$ allows us to apply a generalized Chebyshev inequality (Lemma 13) to establish these bounds. $\square$

We now proceed to the full technical proof.

*Proof.* We will assume that $k = n/2$ is an integer. Let $N = |\mathbb{D}^{\text{in}}| = |\mathbb{D}^{\text{out}}|$, and let $\mathbb{D}^{\text{in}} = \{D_1, \ldots, D_N\}$ and $\mathbb{D}^{\text{out}} = \{D'_1, \ldots, D'_N\}$. Define $a_i = \mathcal{A}(D_i)$ for $D_i \in \mathbb{D}^{\text{in}}$ and $b_j = \mathcal{A}(D'_j)$ for $D'_j \in \mathbb{D}^{\text{out}}$. Let $Z$ be a random variable which is uniformly distributed on $\{a_i\} \cup \{b_j\}$. We may assume without loss of generality that $\mathbb{E}Z = 0$. In what follows, $c$, $\alpha$, $\beta$, and $\gamma$ are constants which we will choose later to optimize our bounds. We also make repeated use of the inequalities $1 + x \leq e^x$ for all $x$; $\frac{1}{1+x} \geq 1 - x$ for all $x \geq 0$; and $e^x \leq 1 + 2x$ and $(1 - x)(1 - y) \geq 1 - x - y$ for $0 \leq x, y \leq 1$. Let $X$ have density proportional to $\exp(-\frac{1}{c\sigma}\|X\|)$. The posterior likelihood ratio is given by

$$f(\hat{\theta}) \overset{\text{def}}{=} \frac{\mathbb{P}(\mathcal{D}_{\text{train}} \in \mathbb{D}^{\text{in}} \mid \hat{\theta})}{\mathbb{P}(\mathcal{D}_{\text{train}} \in \mathbb{D}^{\text{out}} \mid \hat{\theta})} = \frac{\sum_{i=1}^{N} \exp(-\frac{1}{c\sigma}\|\hat{\theta} - a_i\|)}{\sum_{j=1}^{N} \exp(-\frac{1}{c\sigma}\|\hat{\theta} - b_j\|)}.$$

We claim that for all $\hat{\theta}$ with $\|\hat{\theta}\| \leq \gamma\sigma c \log c$, $1 - \frac{\eta}{2} \leq f(\hat{\theta}) \leq (1 - \frac{\eta}{2})^{-1}$. First, suppose that $\|\hat{\theta}\| \leq c^\alpha \sigma$. Then we have:

$$f(\hat{\theta}) \geq \frac{\sum_{\|a_i\| \leq c^\alpha \sigma} \exp[-\frac{1}{c\sigma}(\|\hat{\theta}\| + \|a_i\|)]}{N}$$

$$\geq \frac{(1 - \frac{2}{c^{M\alpha}})N \cdot e^{-2c^{\alpha-1}}}{N}$$

$$\geq 1 - 4c^{-\min(M\alpha, 1-\alpha)}. \tag{11}$$

Otherwise, $\|\hat{\theta}\| \geq c^{\alpha}\sigma$. We now have the following chain of inequalities:

$$f(\hat{\theta}) \geq \frac{\sum_{\|a_i\| \leq c^{\alpha}\sigma} e^{-\frac{1}{c\sigma}(\|\hat{\theta}\| + \|a_i\|)}}{\sum_{\|b_j\| \leq c^{\alpha}\sigma} e^{-\frac{1}{c\sigma}(\|\hat{\theta}\| - \|b_j\|)} + \sum_{c^{\alpha}\sigma < \|b_i\| < \|\hat{\theta}\|} e^{-\frac{1}{c\sigma}(\|\hat{\theta}\| - \|b_j\|)} + \sum_{\|b_i\| \geq \|\hat{\theta}\|} e^{-\frac{1}{c\sigma}(\|b_i\| - \|\hat{\theta}\|)}}$$

$$= \frac{\sum_{\|a_i\| \leq c^{\alpha}\sigma} e^{-\frac{1}{c\sigma}\|a_i\|}}{\sum_{\|b_j\| \leq c^{\alpha}\sigma} e^{\frac{1}{c\sigma}\|b_j\|} + \sum_{c^{\alpha}\sigma < \|b_i\| < \|\hat{\theta}\|} e^{\frac{1}{c\sigma}\|b_j\|} + \sum_{\|b_i\| \geq \|\hat{\theta}\|} e^{\frac{1}{c\sigma}(2\|\hat{\theta}\| - \|b_i\|)}}$$

$$\geq \frac{N(1 - \frac{2}{c^{M\alpha}})e^{-c^{\alpha-1}}}{N\left(e^{c^{\alpha-1}} + \frac{2}{c^{M\alpha}}e^{\frac{1}{c\sigma}\|\hat{\theta}\|} + \frac{2\sigma^M}{\|\hat{\theta}\|^M}e^{\frac{1}{c\sigma}\|\hat{\theta}\|}\right)}$$

$$\geq \frac{(1 - \frac{2}{c^{M\alpha}})e^{-c^{\alpha-1}}}{e^{c^{\alpha-1}} + \frac{2}{c^{M\alpha}}e^{\gamma \log c} + \frac{2}{c^{M\alpha}}e^{\gamma \log c}}$$

$$\geq 1 - 2c^{-M\alpha} - c^{\alpha-1} - 2c^{\alpha-1} - 4c^{\gamma-M\alpha} \tag{12}$$

$$\geq 1 - 9c^{-\min(1-\alpha, M\alpha-2\gamma)}.$$

Combining this with (11) shows that $f(\hat{\theta}) \geq 1 - 9c^{-\min(1-\alpha, M\alpha-\gamma)}$ for all $\|\hat{\theta}\| \leq \gamma\sigma c \log c$.

Next, we must measure the probability of $\|\hat{\theta}\| \leq \gamma\sigma c \log c$. We can lower bound this probability by first conditioning on the value of $\mathcal{D}_{\text{train}}$:

$$\mathbb{P}(\|\hat{\theta}\| \leq \gamma\sigma c \log c) = \frac{1}{|\mathbb{D}|} \sum_{D \in \mathbb{D}} \mathbb{P}(\|\hat{\theta}\| \leq \gamma\sigma c \log c \mid \mathcal{D}_{\text{train}} = D)$$

$$\geq \frac{1}{|\mathbb{D}|} \sum_{\|\mathcal{A}(D)\| \leq c\sigma} \mathbb{P}(\|X\| \leq \gamma\sigma c \log c - \|\mathcal{A}(D)\|)$$

$$\geq \left(1 - \frac{1}{c^M}\right)\left(1 - \frac{1}{2}\exp\left(-\frac{\gamma\sigma c \log c - c\sigma}{c\sigma}\right)\right)$$

$$= \left(1 - \frac{1}{c^M}\right)\left(1 - \frac{e}{2}c^{-\gamma}\right)$$

$$\geq 1 - c^{-M} - \frac{e}{2}c^{-\gamma}.$$

Note that the exact same logic (reversing the roles of the $a_i$'s and $b_j$'s) shows that $f(\hat{\theta}) \leq (1 - 9c^{-\min(1-\alpha, M\alpha-2\gamma)})^{-1}$ with probability at least $1 - c^{-M} - \frac{e}{2}c^{-\gamma}$ as well.

Finally, we can invoke the result of Proposition 2. Let $\Delta = 9c^{-\min(1-\alpha, M\alpha-\gamma)}$ and note that

$$1 - \Delta \leq f(\hat{\theta}) \leq (1-\Delta)^{-1} \implies \max\left\{\mathbb{P}(\mathbf{x}^* \in \mathcal{D}_{\text{train}} \mid \hat{\theta}), \mathbb{P}(\mathbf{x}^* \notin \mathcal{D}_{\text{train}} \mid \hat{\theta})\right\} \leq \frac{1}{2} + \frac{\Delta}{2}.$$

Thus we have

$$
\int \max \left\{ \mathbb{P}(\mathbf{x}^* \in \mathcal{D}_{\text{train}} \mid \hat{\theta}), \mathbb{P}(\mathbf{x}^* \notin \mathcal{D}_{\text{train}} \mid \hat{\theta}) \right\} d\mathbb{P}(\hat{\theta})
$$

$$
\leq \left( \frac{1}{2} + \frac{\Delta}{2} \right) \mathbb{P}(f(\hat{\theta}) \in [1 - \Delta, (1 - \Delta)^{-1}]) + \mathbb{P}(f(\hat{\theta}) \notin [1 - \Delta, (1 - \Delta)^{-1}])
$$

$$
\leq \frac{1}{2} + \frac{9}{2} c^{-\min(1 - \alpha, M\alpha - \gamma)} + 2c^{-M} + ec^{-\gamma}
$$

$$
\leq \frac{1}{2} + \left( \frac{9}{2} + e \right) c^{-\min(1 - \alpha, M\alpha - \gamma, \gamma)} + 2c^{-M} \tag{13}
$$

$$
\leq \frac{1}{2} + 7.5 c^{-\frac{M}{M+2}}
$$

where the last inequality follows by setting $\gamma = 1 - \alpha = M\alpha - \gamma$ and solving, yielding $\gamma = M/(M + 2)$. Solving for $\eta = 7.5 c^{-\frac{M}{M+2}}$, we find that $c = (\frac{7.5}{\eta})^{1+2/M}$ suffices. This completes the proof. $\qquad\square$

**Corollary 10.** *Let $M \geq 2$, $\sigma_i^M \geq \mathbb{E}|\theta_i - \mathbb{E}\theta_i|^M$, and define $\|x\|_{\sigma,M} = \left( \sum_{i=1}^d \frac{|x_i|^M}{d\sigma_i^M} \right)^{1/M}$ for any vector $x = (x_1, \ldots, x_d)^\top$. Generate $Y_i \sim \text{GenNormal}(0, \sigma_i, M)$, $U = Y/\|Y\|_{\sigma,M}$ and draw $r \sim$ Laplace $\left((6.16/\eta)^{1+2/M}\right)$. Finally, set $X = rU$ and return $\hat{\theta} = \theta + X$. Then $\hat{\theta}$ is $\eta$-MIP.*

*Proof sketch.* Let $\|\cdot\| = \|\cdot\|_{\sigma,M}$. It can be shown that the density of $X$ has the proper form. Furthermore, by definition of the $\sigma_i$, we have $\mathbb{E}\|\theta - \mathbb{E}\theta\|^M \leq 1$. The corollary follows directly from Theorem 9. The improvement in the numerical constant (from 7.5 to 6.16) comes from numerically optimizing some of the bounds in Theorem 9, and these optimizations are valid for $M \geq 2$. $\qquad\square$

**Corollary 14.** *When $M \geq 2$, taking $c = (6.16/\eta)^{1+2/M}$ guarantees $\eta$-MIP.*

*Proof.* The constant improves as $M$ increases, so it suffices to consider $M = 2$. Let $M = 2$ and $\alpha = \gamma = 1/2$, and refer to the proof of Theorem 9. Equation (12) can be improved to

$$
1 - 2c^{-M\alpha} - c^{\alpha-1} - (e-1)c^{\alpha-1} - 4c^{\gamma-M\alpha} = 1 - (4 + e + 2c^{-1/2})c^{-1/2}
$$

using the inequality $e^x \leq 1 + (e-1)x$ for $0 \leq x \leq 1$ instead of $e^x \leq 1 + 2x$, which was used to prove Theorem 9. With $\Delta = (4 + e + 2c^{-1/2})c^{-1/2}$, (13) becomes

$$
\frac{1}{2} + \left( \frac{4 + e + 2c^{-1/2}}{2} + e + 2c^{-3/2} \right) c^{-1/2}. \tag{14}
$$

Observe that since $\eta \leq 1/2$, when we set $c = (6.16/\eta)^2$, we always have $c \geq (2 \cdot 6.16)^2$, in which case

$$
\frac{4 + e + 2c^{-1/2}}{2} + e + 2c^{-3/2} \leq 6.1597.
$$

Thus, with $c = (6.16/\eta)^2$, we have

$$
(14) \leq \frac{1}{2} + 6.1597 \cdot \frac{\eta}{6.16} \leq \frac{1}{2} + \eta.
$$

This completes the proof. $\qquad\square$

**Corollary 10.** *Let $M \geq 2$, $\sigma_i^M \geq \mathbb{E}|\theta_i - \mathbb{E}\theta_i|^M$, and define $\|x\|_{\sigma,M} = \left( \sum_{i=1}^d \frac{|x_i|^M}{d\sigma_i^M} \right)^{1/M}$ for any vector $x = (x_1, \ldots, x_d)^\top$. Generate $Y_i \sim \text{GenNormal}(0, \sigma_i, M)$, $U = Y/\|Y\|_{\sigma,M}$ and draw $r \sim$ Laplace $\left((6.16/\eta)^{1+2/M}\right)$. Finally, set $X = rU$ and return $\hat{\theta} = \theta + X$. Then $\hat{\theta}$ is $\eta$-MIP.*

*Proof.* We with to apply the result of Theorem 9 with $\|\cdot\| = \|\cdot\|_{\sigma,M}$. To do this, we must bound the resulting $\sigma^M$ and show that the density of $X$ has the correct form. First, observe that

$$\sigma^M = \mathbb{E}\|\theta - \mathbb{E}\theta\|^M = \sum_{i=1}^{d} \mathbb{E}\frac{|\theta_i - \mathbb{E}\theta_i|^M}{d\sigma_i^M} \leq \sum_{i=1}^{d} \frac{1}{d} = 1.$$

It remains to show that the density has the correct form, i.e. depends on $X$ only through $\|X\|$. This will be the case if the marginal density of $U$ is uniform. Let $p(U)$ be the density of $U$. Observe that, for any $\|u\| = \|u\|_{s,M} = 1$, we have that $Y \mapsto u$ iff $Y = su$ for some $s > 0$. Thus

$$p(u) \propto \int_{s=0}^{\infty} e^{\frac{1}{\sigma_1^2}(su_1/\sigma_1)^M + \cdots + (su_d/\sigma_d)^M} \, ds$$

$$= \int_{s=0}^{\infty} e^{-s^M d\|u\|^2} \, ds$$

$$= \int_{s=0}^{\infty} e^{-s^M d} \, ds.$$

The last inequality holds because $\|u\| = 1$ is constant. Thus, the density is independent of $u$ and we can directly apply Theorem 9. $\qquad\square$

**Proposition 11.** *For any finite $D \subseteq \mathbb{R}$, define $\mathcal{A}(D) = \frac{1}{\sum_{x \in D} x}$. Given a dataset $\mathcal{D}$ of size $n$, define $\mathbb{D} = \{D \subseteq \mathcal{D} : |D| = \lfloor n/2 \rfloor\}$, and define*

$$\sigma^2 = \mathrm{Var}(\mathcal{A}(D)), \qquad \Delta = \max_{D \sim D' \in \mathbb{D}} |\mathcal{A}(D) - \mathcal{A}(D')|.$$

*Here the variance is taken over $D \sim \mathrm{Unif}(\mathbb{D})$. Then for all $n$, there exists a dataset $\mathcal{D}$ with $|\mathcal{D}| = n$ such that $\sigma^2 = O(1)$ but $\Delta = \Omega(2^{n/3})$.*

*Proof.* Assume $n$ is even for simplicity. Let $p = \binom{n}{n/2}^{-1}$ and $A = \sqrt{p} - \sum_{i=0}^{\frac{n}{2}-2} 2^i$. Take

$$\mathcal{D} = \{2^i : i = 0, \ldots, n-2\} \cup \{A\}.$$

We claim that this dataset satisfies the conditions of the proposition.

First, we bound $\sigma^2$. Let $D^* = \{2^0, \ldots, 2^{\frac{n}{2}-2}, A\}$. Note that $\mathcal{A}(D^*) = p^{-1/2}$, and this occurs with probability $p$. For all other subsets $D' \neq D^*$, $0 \leq \mathcal{A}(D') \leq 1$. From this, it can be seen that $0 \leq \mathbb{E}[\mathcal{A}(D)] \leq 2$. The lower bound is trivial since $\mathcal{A}(D) \geq 0$ for all $D$. For the upper bound, we have that

$$\mathbb{E}[\mathcal{A}(D)] = p \cdot \mathcal{A}(D^*) + \sum_{D' \neq D^*} p \cdot \mathcal{A}(D')$$

$$\leq p^{1/2} + \left(\binom{n}{n/2} - 1\right) \cdot p \cdot 1$$

$$= p^{1/2} + 1 - p$$

$$\leq 2,$$

where the equality in the second to last step holds by definition of $p$. We can now use this to bound $\sigma^2$. Since $0 \leq \mathcal{A}(D') \leq 1$ for all $D' \neq D^*$, we have $(\mathcal{A}(D') - \mathbb{E}[\mathcal{A}(D)])^2 \leq 4$ for all $D' \neq D^*$. It follows that

$$\sigma^2 = p \cdot (\mathcal{A}(D^*) - \mathbb{E}[\mathcal{A}(D)])^2 + \sum_{D' \neq D^*} p \cdot (\mathcal{A}(D') - \mathbb{E}[\mathcal{A}(D)])^2$$

$$\leq p \cdot (p^{-1/2} - 0)^2 + \left(\binom{n}{n/2} - 1\right) \cdot p \cdot 4$$

$$= 1 + 4 - 4p$$

$$\leq 5.$$

Thus $\sigma^2 = O(1)$ as desired.

It remains to lower bound $\Delta$. Let $D' = \{2^0, \ldots, 2^{\frac{n}{2}-1}\}$ and note that $D' \sim D^*$. Observe that

$$\begin{aligned}
|\mathcal{A}(D^*) - \mathcal{A}(D')| &\geq p^{-1/2} - 1 \\
&\approx \left( \sqrt{\frac{2}{\pi n}} \cdot 2^n \right)^{1/2} - 1 \\
&\geq 2^{n/3}
\end{aligned} \tag{15}$$

for $n$ large enough, where we have used Stirling's approximation for factorials in equation (15) and $\approx$ means that asymptotically the ratio of the two expressions approaches 1. This completes the proof. $\qquad\square$

