# OpenReview forum: "Provable Membership Inference Privacy"
_TMLR — Accepted by TMLR_

### Review · Reviewer_VuDy · 2023-12-16

**Summary Of Contributions:**

This paper defines a new privacy notion, which is the membership inference privacy (MIP). MIP is a notion of privacy where an adversary is not able to infer membership of a data point in the used dataset with better confidence than random guessing.
MIP is then compared to differential privacy (DP) and a relationship between the two is established and quantified. The required randomness for an algorithm to be MIP is then studied and it is shown that in some cases this randomness is less than what is required for DP.
The paper then proposes a method to make an algorithm with continuous outputs satisfy MIP.

**Audience:**

Yes

**Broader Impact Concerns:**

No concerns of ethical implications detected.

**Claims And Evidence:**

Yes

**Requested Changes:**

- The issues stated in the weaknesses section need to be clarified, and those are critical to recommend acceptance of the paper.
- At this point I am not very convinced with the usefulness of this privacy notion. Moreover, the statements, theorems and proofs need to be clarified. Stronger examples showing the significance of this notion would be very important in motivating this problem better.
- More experiments would benefit this paper greatly, especially in terms of privacy leaks as compared to DP. One experiment that could be useful is assessing privacy leakage with a DP algorithm and a MIP algorithm for the same accuracy (or for the same $\epsilon$ ($\eta$)). This would help to see how it compares with DP and how much sacrifice in terms of privacy one needs to make.
- Many statements are not very clear in the text, so proofreading is necessary to help in the understanding of the readers.

**Strengths And Weaknesses:**

Strengths:
- The paper describes a notion of privacy that makes it easier to understand and audit privacy measures especially empirically.
- The work is interesting and mostly sound.
- The experiments show that MIP improves on DP in terms of percentage of error.

Weakness:
- The MIP notion of privacy has better accuracy, but this is not surprising since the privacy guarantee is weaker and the randomness added is lower. It seems like there is a need for better comparison with DP with respect to privacy breach as well to have a fairer comparison.
- It is not very clear to me how useful this privacy notion is and where it would make sense to allow for a $much$ more lenient privacy for the sake of better accuracy when releasing a model or when releasing data.
- The sentence in section 2, "In particular, in the case of membership inference, one must consider $all$ datasets..." - this seems to imply that MIP is stronger than DP, while this is not the case.
- In theorem 3, it seems like the result is found by defining $I_A(x^*,A(D_{train}))=I_f(x^*,f(A(D_{train})))$, but why is this statement true?
- In many theorems and theorem proofs, it seems that some relations or expansions are used which are not specified to help in understanding the work. For example, the remark after Theorem 4, that when $\epsilon\approx 0$, we have $\delta+(1-\delta)/(1+e^{-\epsilon})-1/2\approx\delta/2+(1-\delta)(\epsilon/4)$ -- getting to $(1-\delta)\epsilon/4$ is unclear. Same with the next statement when $\delta\approx 0$.
- In Theorem 8, "must have $TPR< FPR+\eta$", should this be $\leq$?

---

> ### Author Response · Authors · 2024-02-22
>
> >The MIP notion of privacy has better accuracy, but this is not surprising since the privacy guarantee is weaker and the randomness added is lower. It seems like there is a need for better comparison with DP with respect to privacy breach as well to have a fairer comparison.
>
>
> There is an inherent tradeoff between the level of privacy and the amount of noise added to an algorithm’s output; it is impossible to guarantee more privacy with less noise simply by introducing a new definition. Our goal with MIP is explicitly to introduce a notion of privacy which is weaker than DP (see the third bullet point at the end of Section 1), but which nevertheless still offers some guarantees of privacy.
>
>
> >It is not very clear to me how useful this privacy notion is and where it would make sense to allow for a  more lenient privacy for the sake of better accuracy when releasing a model or when releasing data.
>
>
> We believe that in many "low stakes" scenarios where privacy is nevertheless desired, weaker privacy notions such as MIP may be appropriate. For instance, in recommender systems/online advertising placement, user data may be less sensitive than patient data in a healthcare setting or client data in finance. These decisions should be made on a case-by-case basis; the point of our work is to expand the range of options available to practitioners, rather than focusing on specific applications. Since the guarantees provided by MIP are easily interpretable, it will also be easy for a practitioner to decide when it is appropriate. We have added this discussion to Section 6 of the paper.
>
>
> >The sentence in section 2, "In particular, in the case of membership inference, one must consider all datasets..." - this seems to imply that MIP is stronger than DP, while this is not the case.
>
>
> The purpose of this sentence was to emphasize that our results showing that DP implies MIP are not trivial extensions of the "operational" definition of DP. The need to distinguish between two *sets* of subsets rather than two *individual* subsets complicates the problem from a mathematical perspective. We've added this discussion to the paper after the statement of Theorem 4 to prevent confusion.
>
>
> >In theorem 3, it seems like the result is found by defining $I_A(x^*, A(D_{train})) = I_f(x^*, f(A(D_{train})))$, but why is this statement true?
>
>
> This equality is the definition of the membership inference attack $I_A$. In more detail, a membership inference attack against $A$ can be *any* binary function of the attacked point $x^*$ and the algorithm output $A(D_{train})$. $I_f(x^*, f(A(D_{train})))$ is such a function, so it is permissible to define $I_A$ in this manner. Since MIP guarantees apply to *any* membership inference attack, we get the desired inequality. We've added this clarification to the paper in the proof of Theorem 3.
>
>
> >In many theorems and theorem proofs, it seems that some relations or expansions are used which are not specified to help in understanding the work. For example, the remark after Theorem 4, that when $\epsilon\approx 0$, we have $\delta + (1-\delta)/(1+e^{-\epsilon}) - 1/2 \approx \delta/2 + (1-\delta)(\epsilon/4)$ -- getting to $(1-\delta)(\epsilon/4)$ is unclear. Same with the next statement when $\delta \approx 0$.
>
>
> The first approximate equality is derived from the Taylor expansion of $1/(1+e^{-\epsilon})$. In particular, we have $1/(1+e^{-\epsilon}) = 1/2 + \epsilon/4 + O(\epsilon^2) \approx 1/2 + \epsilon/4$ when $\epsilon \approx 0$, leading to the first approximation. The second statement that $\eta \approx \epsilon/4$ follows by setting $\delta \approx 0$ in the approximate equality $\eta \approx \delta/2 + (1-\delta)\epsilon/4$. We have added these details to the paper after the statement of Theorem 4.
>
>
> >In Theorem 8, "must have $TPR < FPR + \eta$", should this be $\leq$?
>
>
> You are correct, this was a typo. We have fixed this typo.
>
>
> >At this point I am not very convinced with the usefulness of this privacy notion. Moreover, the statements, theorems and proofs need to be clarified. Stronger examples showing the significance of this notion would be very important in motivating this problem better.
>
>
> We hope that the clarifications to your specific questions on some of the proofs above are sufficient; if some statements are still unclear, we would be happy to clarify these as well. Regarding usefulness, we believe that expanding the toolkit available to practitioners can be useful especially in "lower stakes" settings (e.g., recommender systems or personalized advertising, where very strict privacy enforcement is not necessary). Because of the ease with which MIP can be interpreted, it will be easy for a practitioner to decide whether or not it is appropriate on a case-by-case basis. The theoretical results showing that the gap in utility when applying DP vs. MIP can be arbitrarily large provide further evidence that using MIP can provide significant advantages.

---

> > ### Author Response · Authors · 2024-02-22
> >
> > >More experiments would benefit this paper greatly, especially in terms of privacy leaks as compared to DP. One experiment that could be useful is assessing privacy leakage with a DP algorithm and a MIP algorithm for the same accuracy (or for the same $\epsilon$ ($\eta$)). This would help to see how it compares with DP and how much sacrifice in terms of privacy one needs to make.
> >
> >
> > Fig. 1 shows the change in utility at a fixed attacker advantage using DP and MIP privacy accounting, i.e., for the same $\eta$ as you suggested. Fig. 1 shows that the amount of noise required to be added to the base algorithm output is much lower when using direct MIP accounting, as opposed to computing the noise requirement via converting DP to a membership inference guarantee.  We agree that extensive empirical evaluation is an important direction for future work, and we list this in the "Future Work" section of the paper. The purpose of this paper is to introduce the theoretical underpinnings and properties of MIP, and the numerical simulations are meant to illustrate some of the theoretical results. A full evaluation on real data is out of scope for this paper.
> >
> >
> > >Many statements are not very clear in the text, so proofreading is necessary to help in the understanding of the readers.
> >
> >
> > Thanks for pointing out instances where some of the statements were unclear, and for catching the typo in Theorem 8. To recap, our improvements to readability include the following:
> > - We added a thorough discussion of the differences between MIP and DP as privacy notions which make explicit the advantages and disadvantages of each. We’ve also added Table 1 to the introduction, which gives a high level summary of the two privacy notions.
> > - We added clarification to the claims about the difference in hypothesis testing between MIP and DP in Section 2.
> > - We expanded the proof of Prop. 2 to include steps that were previously omitted.
> > - We added an explanation of the equality used in Theorem 3.
> > - We added an explanation of the remark following Theorem 4.
> >
> >
> > We hope that our added discussions and clarifications have made the paper easier to understand, as well as made our contribution clearer. If there are remaining questions, we are happy to answer them.

---

> > > ### Comment · Reviewer_VuDy · 2024-03-01
> > >
> > > Thanks for the response, this mostly clarifies my concerns.
> > > I believe you misunderstood my first comment. I was asking for something like a privacy/accuracy tradeoff, where it is shown exactly how much the privacy is decreasing (how much information is being leaked) in MIP as compared to DP (possibly versus how much the accuracy is improving). This comment was meant to try to have a better understanding for someone for example who would like to choose which scheme to use for their purposes. Can you clarify if this is something possible to do?

---

> > > > ### Author Response · Authors · 2024-03-06
> > > >
> > > > Thank you for your additional feedback, we're glad to hear we've addressed most of your concerns!
> > > >
> > > > Thanks also for clarifying the question in your first comment. Since privacy is a broader and perhaps more nebulous concept than accuracy, we do not believe it is possible to give a single number/metric which fully captures the notion of privacy decrease/information leakage as you’ve mentioned. Instead, throughout the paper, we discuss several different properties of DP vs. MIP which capture some aspect of privacy leakage. See, for instance, the discussion we’ve added on page 5 after Definition 1: MIP may leak information about the base dataset $\mathcal{D}$, whereas DP should provide more protection against this. On the other hand, our results show that MIP and DP both protect against distinguishing between subsets of $\mathcal{D}$.
> > > >
> > > > We also remark that a common measure of privacy/information leakage in the literature is via membership inference attacks; this was part of our motivation for choosing membership inference as the core quantity for our privacy notion.

---

### Review · Reviewer_zNTL · 2023-12-21

**Summary Of Contributions:**

The paper presents Membership Inference Privacy (MIP) as an alternative privacy definition (to Differential Privacy) that focuses on designing a weaker, but more practical and interpretable privacy technique in the context of Membership Inference Attacks. By designing a privacy mechanism around MIAs the authors claim that significantly lower and non-isotropic noise can used to obtain strong privacy guarantees. Here, the privacy strength is measured as the probability advantage of the adversary in determining the membership of a given target sample given access to only the private algorithm’s output and the target sample.

The paper further discusses the theoretical guarantees of MIP, its practical implications, and the connection between Differential Privacy (DP) and MIP.

**Audience:**

Yes

**Broader Impact Concerns:**

The authors do include a limitations section and I have inquired about a few more weaknesses above. Otherwise, the paper does not seem to have other broader impact concerns.

**Claims And Evidence:**

Yes

**Requested Changes:**

My primary concerns and questions have been presented in the weakness section. Below, I have included additional considerations/questions for the authors,

- Can a simple comparison over the privacy-utility trade-off be made with MIP and vanilla DP (say with advanced composition and/or with subsampling) for a DP-SGD pipeline (by appropriately replacing DP with MIP) for a real dataset like MNIST? Furthermore, can simple MIAs be carried out on such a pipeline?
- Finally, is there a way to get sub-sampling-based privacy amplification with MIP?

**Strengths And Weaknesses:**

### Strengths:

- MIP provides a different kind of privacy interpretability than DP that demonstrates the probabilistic strength of the adversary to identify membership of a target (unlike DP) which might be easier to interpret in the context of MIAs.
- The noise required (as shown in Algorithm 1 and Figure 1) for MIP is considerably lower and can be non-isotropic and thus tailored to each dimension.
- The paper demonstrates a simple relationship between MIP and differential privacy with appropriate proofs and counter-examples.
- Experiments (in Figure 2) show the advantage of MIP over DP for the privacy-utility trade-off.


### Weaknesses:

- Figure 1 is quite confusing. It is unclear what algorithm naive DP refers to. Furthermore, it is unclear why the noise level increases with the number of samples (and not the data dimensions). Normally higher samples should provide better utility-privacy tradeoff. Could the exact naive DP algorithm be added to the paper? Also what composition theorems have been used to compute the privacy of DP in Figure 1? Without comparisons between MIP and composed DP (say with advanced composition) it is hard to know the exact/true benefit extracted from MIP.
- In Figure 2, is MIP compared to the same naive DP algorithm and does such an algorithm have noise scaled with ’n’? It might be more appropriate to compare MIP against DP, DP with advanced composition, and maybe even DP with subsampling-based privacy amplification.
- The data-dependent approach of MIP is an advantage as it then requires lower noise levels. However, such non-private computations can leak private data. Furthermore, can the privacy leakage due to such a data-dependent noise mechanism be quantified or simply compared to regular DP (even for simple toy datasets)?
- Although MIP guarantees privacy in the adversary algorithm setting, do the same benefits extend to other attacks such as linkage attacks (like regular DP)? More generally, can MIP be particularly vulnerable to other privacy attacks due to its data-dependent approach or due to adding lower amounts of noise?
- MIP might be incompatible with local privacy setups or federated setups.

---

> ### Author Response · Authors · 2024-02-22
>
> >Figure 1 is quite confusing. It is unclear what algorithm naive DP refers to. Furthermore, it is unclear why the noise level increases with the number of samples (and not the data dimensions). Normally higher samples should provide better utility-privacy tradeoff. Could the exact naive DP algorithm be added to the paper? Also what composition theorems have been used to compute the privacy of DP in Figure 1? Without comparisons between MIP and composed DP (say with advanced composition) it is hard to know the exact/true benefit extracted from MIP.
>
>
> "Naively applying DP" refers to bounding the attacker advantage $\eta$ based on enforcing $\epsilon$-DP, then using the tight correspondence between $\epsilon$ and $\eta$ from Theorem 4. The algorithm used to enforce DP in this case is the exponential mechanism (see the description in paragraph beginning "Refer to Fig. 1" just under the figure on pg. 9 of the paper). There are no composition results used in this case; we have simply bounded the sensitivity of the algorithm and used standard results for the Laplace mechanism in DP. We have added these details to the paper in the caption for Fig. 1.
>
>
> The reason that the noise level increases with the number of samples can be seen in the proof of Proposition 11. Essentially, as the sample size increases, the sensitivity of the algorithm's output grows arbitrarily large (requiring larger noise for DP), while the variance of the algorithm's output remains bounded by a constant (meaning larger noise is not required for MIP).
>
>
> >In Figure 2, is MIP compared to the same naive DP algorithm and does such an algorithm have noise scaled with ’n’? It might be more appropriate to compare MIP against DP, DP with advanced composition, and maybe even DP with subsampling-based privacy amplification.
>
>
> We have elected to remove this figure to focus entirely on the theoretical aspect of MIP. Please refer to the general comment for a more thorough discussion.
>
>
> >The data-dependent approach of MIP is an advantage as it then requires lower noise levels. However, such non-private computations can leak private data. Furthermore, can the privacy leakage due to such a data-dependent noise mechanism be quantified or simply compared to regular DP (even for simple toy datasets)?
>
>
> We use $\mathcal{D}_t$ to denote D_train; the LaTeX has problems rendering on OpenReview. We discuss this briefly in the “Limitations” section of the paper. Specifically, MIP guarantees specify that the adversary cannot distinguish between points in $\mathcal{D}_t$ and $\mathcal{D} \setminus \mathcal{D}_t$; however, this does not rule out the possibility that the adversary may be able to determine whether or not a point was in the “base” dataset $\mathcal{D}$ or not. Considerations of this sort are essentially the content of Sections 3.2 and 3.3, where we discuss the motivation for this design choice.
>
>
> Regarding quantifying this privacy leakage, the leakage about membership information in $\mathcal{D}$ can be arbitrarily high. Consider, for instance, the following “algorithm”: $\mathcal{A}(D) = \mathbb{I}(D \subseteq \mathcal{D})$. The output of this algorithm gives the adversary no information about whether or not a given $\mathbf{x}^*$ was in $\mathcal{D}_t$ for any random subset of $\mathcal{D}$, so no noise needs to be added to it for it to satisfy MIP. However, the output very clearly reveals information about membership in $\mathcal{D}$. For this reason, in order for MIP to apply, membership in the “overall population” $\mathcal{D}$ should not be considered private, only membership in $\mathcal{D}_t$. We have expanded the discussion in Section 3.3 to make this point clearer.

---

> > ### Author Response · Authors · 2024-02-22
> >
> > >Although MIP guarantees privacy in the adversary algorithm setting, do the same benefits extend to other attacks such as linkage attacks (like regular DP)? More generally, can MIP be particularly vulnerable to other privacy attacks due to its data-dependent approach or due to adding lower amounts of noise?
> >
> >
> > One blanket statement that we can make regarding other privacy attacks is that MIP protects against any privacy attack which would *imply* a successful membership inference attack. One instance of this would be reconstruction attacks, where the adversary seeks to reconstruct training data "ex nihilo" rather than being *supplied* with potentially included data. If an attacker can successfully reconstruct the training data, then a simple membership inference attack is to reconstruct the training data first and check whether or not the attacked record is contained in the reconstruction. If membership inference is impossible by enforcing MIP, then this attack cannot succeed with high probability, so reconstruction cannot succeed with high probability either.
> >
> >
> > For some versions of linkage attacks, MIP may not provide protection. For instance, in a standard version of a linkage attack, the adversary already knows that a specific individual was contained in the training dataset, just not which specific *record* belongs to that individual. Such an attacker has already passed the "line of defense" we are seeking to enforce with MIP (namely, the membership of an individual in the training data), so MIP will not apply here.
> >
> >
> > We have added these discussions to the paper in the “Limitations” paragraph of Section 6.
> >
> >
> > >MIP might be incompatible with local privacy setups or federated setups.
> >
> >
> > We agree that more work is required to extend MIP to these settings. Indeed, there are a number of different possible goals that one may have for such a setting. Should an adversary be unable to tell if a target record is contained in the union of all of the local datasets, or just be unable to tell which specific user a target record has come from? Is some amount of statistical similarity between the users' data required? There are many important design choices for these extensions and we agree that it is an interesting open question, but it is out of scope for the present work.
> >
> >
> > >Can a simple comparison over the privacy-utility trade-off be made with MIP and vanilla DP (say with advanced composition and/or with subsampling) for a DP-SGD pipeline (by appropriately replacing DP with MIP) for a real dataset like MNIST? Furthermore, can simple MIAs be carried out on such a pipeline?
> >
> >
> > We agree that extensive empirical evaluation is an important direction for future work, and we list this in the "Future Work" section of the paper. The purpose of this paper is to introduce the theoretical underpinnings and properties of MIP, and the numerical simulations are meant to illustrate some of the theoretical results. A full evaluation on real data is out of scope for this paper.
> >
> >
> >
> > >Finally, is there a way to get sub-sampling-based privacy amplification with MIP?
> >
> >
> > At present, we don't have results for directly enforcing MIP via subsampling. One can always enforce MIP via combining a DP algorithm with the correspondence between DP and MIP in Theorem 4, but we agree that it is plausible that a more refined analysis of subsampling for MIP (similar to Theorem 9) may exist. We note in our "Future Work" paragraph at the end of Section 6 that such analyses may be possible and of future interest, but we have added specific examples (such as subsampling, composition, and group privacy results) to the discussion.

---

> > > ### Comment · Reviewer_zNTL · 2024-03-07
> > >
> > > Thank you for providing detailed clarification for my questions and adding the limitations of MIP to the paper. For the most part, these comments do address my major concerns. There are several unanswered questions about MIP alongside limited empirical evaluations; however, the paper is well-motivated and useful as an introductory theory paper for MIP.

---

### Review · Reviewer_h3cw · 2024-02-08

**Summary Of Contributions:**

The paper proposes membership inference privacy (MIP), a formal definition of (training data) privacy based on membership inference attacks (MIA). The authors show that MIP is related to differential privacy (DP), but weaker in the sense that DP implies MIP, but the reverse is not true. The authors provide some theoretical and empirical evidence that MIP can improve the privacy-utility trade-off compared to DP while guaranteeing protection against MIAs. In summary, while I find the proposed definition interesting, I also find that the paper requires some fixes and more work before it provides sufficient evidence for the claims presented.

**Audience:**

Yes

**Broader Impact Concerns:**

I have no broader impact concerns

**Claims And Evidence:**

No

**Requested Changes:**

Please address the following questions/suggested changes:

1) Fix the privacy leakage issue in the experiment in Sec 5.2: you cannot choose the clipping threshold for DP based on private information like the actual gradient magnitudes without implementing some additional privacy mechanism to control that leakage. Using the actual median was an error in the original Abadi et al. paper, that has been fixed in the later versions e.g. on arXiv.

2) Please clarify: does estimating $\sigma$ from data weaken the resulting MIP guarantee: how does the estimation budget $B$ in Alg 1 affect MIP? Do inaccuracies in estimating $\sigma$ affect the MIP guarantees?

3) Prop2: the proof should be written out more clearly, for example: jumping from Eq4 to the claim seem to skip plenty of work, I am not sure I understand what you mean by writing $\mathcal P (\mathcal A (D))$ vs ) $\mathcal P (\mathcal A_{\mathcal A} (D))$ vs $\mathcal P (\mathcal A_{\mathcal A, D} (D))$), the result is for continuous values while the proof uses discrete sums.

4) Thm 4: The stated bound for the attackers accuracy is exactly the bound shown previously by Humpries et al. 2023 (please also add Humpries et al. to the Related work; see Thudi et al. 2022 for a discussion on the existing bounds). Also, Thm 4 proof should also explicitly argue that the bound is tight as claimed in the Thm statement. This seems unclear without further argument as the proof relies on several inequalities (we know this is tight under DP up to tightness of the DP guarantees as shown by Thudi et al.2022).

5) Thm 8: should also mention that corresponding result is known for DP already from Yeom et al. 2018.

6) Mention adjancency explicitly in defining DP and MIP, or mention if MIP is agnostic w.r.t. adjacency same as DP. At least proof of Lemma 12 in the Appendix and all the experiments use replace adjacency.

7) Discuss or at least mention if MIP provides group MIP and composition: these are fundamental properties of DP, and it would be important to know if these hold for MIP.


8) Add $M=2$ line in Fig 2a; feel free to use separate zoom plots or simply limit the y-axis to keep the other resultsd readable, but dropping MIP lines in case they are bad seems like cherry picking the results somewhat.

9) Please clarify: p3 in defining DP: "While similar to the setting of membership inference at face value, there are subtle differences. In particular, in the case of membership inference, one must consider all datasets which could have contained the target record, and all datasets which do not contain the target record, and distinguish between the algorithm’s output in this larger space of possibilities. This is in contrast to the characterization of DP above, which only needs to distinguish between two datasets." I do not understand the claim: DP def requires the bound to hold for all possible adjacent data sets. How is this different from MI?


### Typos and other minor changes

i) Fix Appendix A, proof of Prop2 lines leading to and including Eq (3): should imply an upper bound for the original prob., not for diff in sums? Eq (4) x should be xˆ*?

ii) Prop 6 proof beginning: missing A from the divisor. Next line dividend should be mathcal A(D'), not mathcal A(D), and the same in the rest of the proof (also in the divisor for consistency)?

iii) Thm 8 should have \leq instead of strict ineq

iv) Mention what are $x_i$ in Cor 10


#### References (beyond those in the paper):

Humpries et al. 2023: Investigating Membership Inference Attacks under Data Dependencies.

**Strengths And Weaknesses:**

### Strengths

1) The authors relate the proposed MIP to DP, which is currently the best-understood formal privacy definition.

2) Showing that several MIA bounds are achievable without DP under the weaker MIP notion is interesting.

3) The proposed MIP might allow for improved privacy-utility trade-off compared to DP, at least in some cases.


### Weaknesses

1) I do not buy the argument that MIP guarantees are much easier to interpret that DP: DP can be equivalently stated as a guarantee against MIA in iid setting (following Yeom et al. 2018), which allows for a roughly correct interpretation without using $(\epsilon,\delta)$.
Also, as DP, MIP is not fully characterised by a single number (e.g. it assumes some underlying larger dataset, and a sampling scheme that produces the actual training data from this underlying population), so stating the guarantee in a readily readable manner requires sweeping this complexity out of sight. I find the reasoning of MIP as relaxed form of DP much more appealing irrespective of how easily one can describe the guarantees to an outsider.

2) Related to the previous point, while the paper explicitly connects MIP to DP, this discussion should be continued and expanded: several results have obvious connections to MI bounds derived under DP, but this is not usually even mentioned. This seems like a wasted opportunity, as deriving the same bound under MIP shows that DP is not strictly required for that bound (see Required changes for details). Also, whether or not MIP has analogues of some major DP properties, namely, composition and group privacy, is not mentioned at all.

3) All the experiments use only synthetic data, there are no results e.g. using known becnhmark datasets or models.

4) The paper seems generally a bit rushed and unfinished: several proofs seem to end while leaving several steps as an exercise for the reader, the notation could be better, and the experiments need some fixing (see Required changes for details).

---

> ### Author Response · Authors · 2024-02-22
>
> >I do not buy the argument that MIP guarantees are much easier to interpret that DP: DP can be equivalently stated as a guarantee against MIA in iid setting (following Yeom et al. 2018), which allows for a roughly correct interpretation without using $(\epsilon, \delta)$. Also, as DP, MIP is not fully characterised by a single number (e.g. it assumes some underlying larger dataset, and a sampling scheme that produces the actual training data from this underlying population), so stating the guarantee in a readily readable manner requires sweeping this complexity out of sight. I find the reasoning of MIP as relaxed form of DP much more appealing irrespective of how easily one can describe the guarantees to an outsider.
>
>
> Theorem 1 of Yeom et al. (2018) shows that DP implies a bound on MIA in an i.i.d. setting, though this is not a true equivalence as the reverse implication does not necessarily hold. While there is an underlying sampling scheme required to fully specify MIP, we hope that this should be easier for a non-expert to understand than, say, explaining probability densities or laws, or hypothesis testing, as would be required for fully specifying DP. Nevertheless, we are happy to emphasize that the main motivation for MIP is as a relaxation of DP and de-emphasize the ease of interpretation (see the characterization provided in the newly added Table 1).
>
>
> >Related to the previous point, while the paper explicitly connects MIP to DP, this discussion should be continued and expanded: several results have obvious connections to MI bounds derived under DP, but this is not usually even mentioned. This seems like a wasted opportunity, as deriving the same bound under MIP shows that DP is not strictly required for that bound (see Required changes for details). Also, whether or not MIP has analogues of some major DP properties, namely, composition and group privacy, is not mentioned at all.
>
>
> See the discussion below on this point. While some quantities related to membership inference under DP may appear similar to those derived with MIP, there are important technical differences which we have attempted to clarify below. We also agree that some other properties such as composition or group guarantees are important topics for future study and have added reference to this fact in the paper, in the “Future Work” paragraph of Section 6.
>
>
> >All the experiments use only synthetic data, there are no results e.g. using known benchmark datasets or models.
>
>
> We agree that extensive empirical evaluation is an important direction for future work, and we list this in the "Future Work" section of the paper. The purpose of this paper is to introduce the theoretical underpinnings and properties of MIP, and the numerical simulations are meant to illustrate some of the theoretical results. A full evaluation on real data is out of scope for this paper.
>
>
> >Fix the privacy leakage issue in the experiment in Sec 5.2: you cannot choose the clipping threshold for DP based on private information like the actual gradient magnitudes without implementing some additional privacy mechanism to control that leakage. Using the actual median was an error in the original Abadi et al. paper, that has been fixed in the later versions e.g. on arXiv.
>
>
> We have elected to remove this figure and focus entirely on the theoretical aspects of MIP. Please refer to the comment on the $\sigma$ estimation method, and to the general comment to all reviewers, for a more thorough discussion.
>
>
> >Please clarify: does estimating $\sigma$ from data weaken the resulting MIP guarantee: how does the estimation budget $B$ in Alg 1 affect MIP? Do inaccuracies in estimating $\sigma$ affect the MIP guarantees?
>
>
> In order for MIP guarantees to hold, $\sigma$ must be an upper bound on the central moment defined on Theorem 9 or Corollary 10. We can only guarantee that the proposed estimation procedure recovers the correct value for $\sigma$ asymptotically as $B\rightarrow\infty$; technically, for any finite $B$,the estimate of $\sigma$ cannot be guaranteed to be a true upper bound. Thus, to strictly enforce MIP, a true upper bound on $\sigma$ must be known. While this is not desirable, we believe it is still an improvement over the sensitivity bound needed for many DP algorithms (e.g., the Laplace mechanism), since $\sigma$ may be consistently estimated from data, but there is no consistent estimator of the sensitivity.
> We view the $\sigma$ estimation procedure as similar to the "folklore" statement in the DP literature that $\epsilon\leq1$ is needed for theory, but $\epsilon\leq10$ is fine in practice. (See e.g. Ponomareva et al. (2023) https://arxiv.org/pdf/2303.00654.pdf, Section 5.2.1.) We've added this caveat to the paper after the statement of Corollary 10.

---

> > ### Author Response · Authors · 2024-02-22
> >
> > >Prop2: the proof should be written out more clearly, for example: jumping from Eq4 to the claim seem to skip plenty of work, I am not sure I understand what you mean by writing  vs )  vs ), the result is for continuous values while the proof uses discrete sums.
> >
> >
> > We have added the calculations required to go from Equation (4) to the result. The subscripts on $\mathbb{P}$ were intended to indicate the source of randomness with respect to which the probability is calculated, but we agree that this notation is unnecessary. We have simplified the notation to make the proof easier to follow.
> >
> >
> > >Thm 4: The stated bound for the attackers accuracy is exactly the bound shown previously by Humpries et al. 2023 (please also add Humpries et al. to the Related work; see Thudi et al. 2022 for a discussion on the existing bounds). Also, Thm 4 proof should also explicitly argue that the bound is tight as claimed in the Thm statement. This seems unclear without further argument as the proof relies on several inequalities (we know this is tight under DP up to tightness of the DP guarantees as shown by Thudi et al.2022).
> >
> >
> > Thank you for pointing out the oversight on the tightness result; we have added an example of an $(\epsilon, \delta)$-DP algorithm where the attacker's accuracy matches the upper bound to the proof of Theorem 4, thus proving tightness.
> >
> >
> > We have added the work of Humphries et al. to the related work. Theorem III.1 of Humpries et al. is a bound on the attacker's accuracy for distinguishing only between two datasets $S = \tilde{S}\cup \{z\}$ and $S'=\tilde{S}\cup\{z'\}$, and the bound follows directly from results of Kairouz et al. In the setting of MIP, the adversary has a more complicated task: it must determine whether the training set was one of the $\binom{n}{n/2}$ sets which contain the target record $\mathbf{x}^*$, or one of the $\binom{n}{n/2}$ sets which do not contain $\mathbf{x}^*$. Showing that bounds on the former scenario imply bounds on the latter is nontrivial, as can be seen from the proof of Theorem 4. See the further discussion on this point below.
> >
> >
> > >Thm 8: should also mention that corresponding result is known for DP already from Yeom et al. 2018.
> >
> >
> > We assume you are referring to Theorem 1 of "Privacy Risk in Machine Learning: Analyzing the Connection to Overfitting" (Yeom et al., 2018), which bounds the difference between an adversary's TPR and FPR in their membership inference setting with i.i.d. data. While the goal of this result is similar to our result in Theorem 8, there are a few differences. First, their membership inference setup is different from ours. Second, their bound is weaker: $e^\epsilon - 1 \rightarrow\infty$, and indeed their bound is vacuous for $\epsilon \geq \log 2$. On the other hand, our upper bound always remains bounded by 1, and even gives a stronger result for DP if $\eta$ is replaced by the corresponding DP parameters from our Theorem 4. We have added the citation to Yeom et al. after Theorem 8, as well as an explanation of the differences between our results and theirs.
> >
> >
> > >Mention adjancency explicitly in defining DP and MIP, or mention if MIP is agnostic w.r.t. adjacency same as DP. At least proof of Lemma 12 in the Appendix and all the experiments use replace adjacency.
> >
> >
> > We have added an explicit definition of adjacency to the paper in the DP paragraph of Section 2. You are correct that we were specifically using replacement adjacency/bounded DP.
> >
> >
> > >Discuss or at least mention if MIP provides group MIP and composition: these are fundamental properties of DP, and it would be important to know if these hold for MIP.
> >
> >
> > We have expanded the "Future Work" section to include discussion of these specific properties. At present, we do not have results for group or composition properties, but we agree that these are important topics for future study.
> >
> >
> > >Please clarify: p3 in defining DP:
> >
> >
> > DP does indeed require the bound to hold for any two datasets. However, the hypothesis testing problem only considers two adjacent datasets *at any given time*. That is, the DP setup is: given $D_0 \sim D_1$, distinguish between these two datasets. The MIP setup is: there are $\binom{n}{n/2}$ datasets containing $\mathbf{x}^*$ and $\binom{n}{n/2}$ datasets which do not contain $\mathbf{x}^*$; distinguish between these two large *collections* of datasets. We emphasized this point because it is nontrivial to show that distinguishing between any two datasets at once also allows one to distinguish between two collections of datasets, as the proof of Theorem 4 shows. We’ve added this clarification after the statement of Theorem 4. We hope this clarifies the distinction, but we are happy to provide further clarification if it is still unclear.
> >
> >
> > >Typos & minor changes.
> >
> >
> > Thank you for pointing these out, we have fixed them.

---

### Author Response · Authors · 2024-02-22

We thank the reviewers for their time and for their detailed and constructive comments. We believe that addressing the issues they raised has improved the clarity of the paper and strengthened it considerably. All of the changes we have made in response to the reviewers’ questions have been indicated in blue text in the revised paper which we have uploaded, and we have tried to indicate the location of changes corresponding to specific questions in our responses.


We wish to make a general comment regarding the TMLR acceptance criteria. Specifically, we agree with the reviewers that the experiments are insufficient to show the advantages of MIP in real-world empirical settings. We have adjusted our claims and restructured the paper to emphasize that the present work is an entirely theoretical contribution. Our purpose is to introduce the concept of MIP itself, analyze the mathematical properties which follow from the definition, and examine the connections and distinctions between MIP and DP. A secondary contribution is the introduction of Algorithm 1. Rather than presenting this as an algorithm for obtaining improved performance on real-world tasks, we propose it as a proof of concept that by considering relaxed notions of privacy, we can, in theory, obtain improved utility as opposed to enforcing DP.


In keeping with this revised presentation of our contributions, we have also elected to remove the short simulation experiment in Section 5.2. We believe this creates the most coherent motivation for the paper, in the sense that all of the results now contained in the paper are strictly motivated by theory. The results previously contained in Section 5.2, specifically Fig. 2, used an approximation to MIP by estimating the values for $\sigma_i$ from data, and therefore do not come with strict theoretical guarantees. While we believe such an approach should be reasonable in practice, thoroughly evaluating and justifying this claim would require expanding the scope of the paper significantly, which we believe is better left for future work.


We hope that by reducing the breadth of our claims, our revised presentation meets the TMLR criterion that there is no gap between the stated results and the provided evidence in the paper. We are open to further discussion if the reviewers believe there is still a gap.


Regarding the second criterion, we are pleased that there is no disagreement among the reviewers that our contribution would be of interest to some members of TMLR’s audience:


- Reviewer h3cw: Privacy-utility trade-off, connection to DP. "Showing that several MIA bounds are achievable without DP under the weaker MIP notion is interesting."


- Reviewer zNTL: Ease of interpretability, lower and adaptive (non-isotropic) noise requirement in some cases, clear relationship between DP and MIP.


- Reviewer VuDy: Ease of interpretability and auditing. "The work is interesting."


In particular, it does not appear that any of the reviewers’ positive assessment of the relevance criterion hinged on the brief results in Section 5.2, so we believe that the more cohesive, entirely theoretical revision of the paper will still pass this bar.

---

### Comment · Reviewer_h3cw · 2024-02-29
**Some further comments**

Thank you for the rebuttal, as a general comment, I think the fixed proofs are much clearer, and the added explanations help to make the paper more readable. Some additional comments/questions:

1) Seems like you have (by accident?) omitted the proof of Prop 11 from the current version.

2) I think some parts (most clearly the abstract, Table 1) still oversell the contributions a bit: " We give a precise characterization of the relationship between MIP and DP, and show that MIP can be achieved using less amount of randomness compared to the amount required for guaranteeing DP, leading to smaller drop in utility". Compare this to the list of contributions, which makes it clear that smaller noise might happen in some situations, it is not a general property of MIP.

3) On Thm8: what I meant with the comment was that the bound on the MI adversary given as TPR-FPR $\leq \eta$, where $\eta$ is the adversary accuracy, was introduced by Yeom et al. (while the specific value for the bound has been improved after that, with the bound given by Thudy et al. being tight for ADP). The main point was that MIP matches what we get with DP, so should point the connection out explicitly. In case you agree, please fix the relevant comment after Thm 8 to reflect this.

* Proof of Prop2, p15 first line should have geq sum over $D \in \mathbb D^{out}$?

---

> ### Author Response · Authors · 2024-03-06
>
> Thank you for the additional feedback! We are glad you are happy with the fixed proofs and added explanations. All of the updates to the manuscript from your further comments are highlighted in the revised version in red, to distinguish them from the first round of revisions.
>
> >Seems like you have (by accident?) omitted the proof of Prop 11 from the current version.
>
> There is a proof in Appendix D, but it is short so we have moved it to the main text of the paper for clarity. Thanks for pointing this out.
>
> >I think some parts (most clearly the abstract, Table 1) still oversell the contributions a bit: " We give a precise characterization of the relationship between MIP and DP, and show that MIP can be achieved using less amount of randomness compared to the amount required for guaranteeing DP, leading to smaller drop in utility". Compare this to the list of contributions, which makes it clear that smaller noise might happen in some situations, it is not a general property of MIP.
>
> We have updated the language in the abstract and in Table 1 to be consistent with the revised list of contributions.
>
> >On Thm8: what I meant with the comment was that the bound on the MI adversary given as TPR-FPR $\leq \eta$, where $\eta$ is the adversary accuracy, was introduced by Yeom et al. (while the specific value for the bound has been improved after that, with the bound given by Thudy et al. being tight for ADP). The main point was that MIP matches what we get with DP, so should point the connection out explicitly. In case you agree, please fix the relevant comment after Thm 8 to reflect this.
>
> Thanks for clarifying this point. We’ve fixed the comment to explicitly point out this connection as follows:
> The connection between the attacker’s accuracy and theTPR/FPR gap is not specific to MIP, and indeed the same bound on this gap in terms of the attacker’s accuracy was shown by Yeom et al. (2018). However, while the form of our bounds is the same, note that Theorem 8 does not follow from Yeom et al. (2018) because the membership inference setting considered in their work is different from ours. (In particular, they focused on membership inference in the presence of i.i.d.data.)
>
> >Proof of Prop2, p15 first line should have geq sum over $D \in \mathbb{D}^{out}$?
>
> Thanks for catching this, we’ve fixed it (and the same error in the denominator of equation (4)).

---

> > ### Comment · Reviewer_h3cw · 2024-03-07
> > **No further question**
> >
> > Thank you for fixing the issues, I have no further questions on this.

---

### Comment · Reviewer_VuDy · 2024-03-07
**No further comments**

I thank the authors for their feedback, I have no further questions.

---

### Decision · Action_Editor_crSP · 2024-03-18

**Recommendation:** Accept with minor revision

**Comment:**

In general the paper is in a good shape. However, I would like the authors to clarify and fix the following points.
- The proof for Proposition 11 needs further clarification. From the current presentation it is difficult to see if the claims on $\sigma^2$ and $\Delta$ hold. Please make the proof connect explicitly to the claim.
- I guess the data set you consider for Fig. 1 is the same that was used in the proof of Prop. 11? If not, please clarify which data set was used, and what was the sensitivity.
- In the discussion after Thm 4, you could mention that it holds for $(\epsilon, \delta)$-DP algorithms $\mathcal{A}$
- Just before Corollary 10, I guess the $x - \mu$ should be in absolute value?

**Audience:**

All the reviewers, as well as I, found the MIP an interesting and novel notion of privacy. Since the practical privacy guarantees of ML methods are an important topic of research, I believe the paper is interesting for the TMLR audience in general.

**Claims And Evidence:**

This paper proposes a new notion for privacy, the membership inference privacy (MIP), which is strictly weaker privacy notion than differential privacy (DP).

The main claim of the paper is that the MIP notion provides meaningful and measurable privacy guarantees against membership inference attacks while allowing adding less noise to the inference compared to DP. Authors show two things to support this claim. First, Theorem 4 shows that any approximate DP algorithm also satisfies MIP, thus noise level satisfying DP is sufficient for MIP. Second, Proposition 11 demonstrates a case where the added noise to meet MIP can be substantially smaller than noise to satisfy DP. This is further illustrated empirically in a simulated data setting for a particular loss function.

The paper underwent a significant revision after the initial reviews, and all the reviewers were satisfied with the changes.

---

> ### Author Response · Authors · 2024-03-27
>
> Thanks again to all of the reviewers and the editor for their helpful input. We are very pleased with the decision! Below we have addressed the final additional comments, and the associated revisions are shown in the updated paper in teal text.
>
> >The proof for Proposition 11 needs further clarification. From the current presentation it is difficult to see if the claims on $\sigma^2$ and $\Delta$ hold. Please make the proof connect explicitly to the claim.
>
> We’ve added the calculations for $\sigma^2$ and $\Delta$ to clarify how the claim follows from the proof. Since the proof now contains several lines of calculations, we have moved it out of the main body and into the appendix (specifically, the end of Appendix D on the final page of the paper) with the rest of the proofs.
>
>
> >I guess the data set you consider for Fig. 1 is the same that was used in the proof of Prop. 11? If not, please clarify which data set was used, and what was the sensitivity.
>
> You are correct, the data set for Fig. 1 is the same as that used for Prop. 11. The discussion after the proof of Prop. 11 mentions this, but we have also added the fact that we’re using this dataset to the caption on Fig. 1 to make it clearer.
>
>
> >In the discussion after Thm 4, you could mention that it holds for $(\epsilon, \delta)$-DP algorithms $\mathcal{A}$.
>
> We’ve updated the discussion to note that the hypothesis testing formulation described here is equivalent to $(\epsilon, \delta)$-DP algorithms $\mathcal{A}$.
>
>
> >Just before Corollary 10, I guess the $x - \mu$ should be in absolute value?
>
> You are correct. Thanks for catching this, we have fixed it.

---

> > ### Comment · Action_Editor_crSP · 2024-03-28
> >
> > Authors have now addressed my remaining comments/concerns.